# Efficiently Vectorized MCMC on Modern Accelerators

**Hugh Dance** [1]  **Pierre Glaser** [1]  **Peter Orbanz** [1]  **Ryan Adams** [2]

## Abstract

With the advent of automatic vectorization tools (e.g., JAX's `vmap`), writing multi-chain MCMC algorithms is often now as simple as invoking those tools on single-chain code. Whilst convenient, for various MCMC algorithms this results in a synchronization problem—loosely speaking, at each iteration all chains running in parallel must wait until the last chain has finished drawing its sample. In this work, we show how to design single-chain MCMC algorithms in a way that avoids synchronization overheads when vectorizing with tools like `vmap`, by using the framework of finite state machines (FSMs). Using a simplified model, we derive an exact theoretical form of the obtainable speed-ups using our approach, and use it to make principled recommendations for optimal algorithm design. We implement several popular MCMC algorithms as FSMs, including Elliptical Slice Sampling, HMC-NUTS, and Delayed Rejection, demonstrating speed-ups of up to an order of magnitude in experiments.

## 1. Introduction

Automatic vectorization is the act of transforming a function to handle batches of inputs without user intervention. Implementations of automatic vectorization algorithms are now available in many mainstream scientific computing libraries, and have dramatically simplified the task of running multiple instances of a single algorithm concurrently. They are routinely used to train neural networks (Flax, 2023) and in other scientific applications, e.g., Schoenholz and Cubuk (2021); Oktay et al. (2023); Pfau et al. (2020).

This paper focuses on the use of automatic vectorization for Markov chain Monte Carlo (MCMC) algorithms. Tools like JAX's `vmap` provide a convenient way to run multiple MCMC chains in parallel: one can simply write single-chain MCMC code, and call `vmap` to turn it into vectorized, multi-chain code that can run in parallel on the same processor. Many state-of-the-art MCMC libraries have consequently adopted machine learning frameworks with automatic vectorization tools as their backend.

One limitation with automatic vectorization tools is how they handle control flow. Since all instructions must be executed in lock-step, if the algorithm has a while loop, all chains must wait until the last chain has finished its iterations. This can lead to large inefficiencies for MCMC algorithms that generate each sample using variable-length while loops. Roughly speaking, if vectorization executes, say, 100 chains in parallel, all but one finish after at most 10 steps, and the remaining chain runs for 1000 steps, then about 99% of the GPU capacity assigned to `vmap` is wasted (and our simulations show the effect can indeed be this drastic). For the No-U-Turn Sampler (HMC-NUTS) (Hoffman et al., 2014), this problem is well-documented (BlackJax, 2019; Sountsov et al., 2024; Radul et al., 2020). It also affects various other algorithms, such as variants of slice sampling (Neal, 2003; Murray et al., 2010; Cabezas and Nemeth, 2023), delayed rejection methods (Mira et al., 2001; Modi et al., 2024) and unbiased Gibbs sampling (Qiu et al., 2019).

In this work, we show how to transform MCMC algorithms into equivalent samplers that avoid these synchronization barriers when using `vmap`-style vectorization. In particular,

1. We develop a novel approach to transform MCMC algorithms into finite state machines (FSMs), that can avoid synchronization barriers with `vmap`.

2. We analyze the time complexity of our FSMs against standard MCMC implementations and derive a theoretical bound on the speed-up under a simplified model.

3. We use our analysis to develop principled recommendations for optimal FSM design, which enable us to nearly obtain the theoretical bound in speed-ups for certain MCMC algorithms.

4. We implement popular MCMC algorithms as FSMs, including Elliptical Slice Sampling, HMC-NUTS, and Delayed-Rejection MH - demonstrating speed-ups of up to an order of magnitude in experiments.

---

[1]Gatsby Unit, University College London, UK [2]Department of Computer Science, Princeton University, USA. Correspondence to: Hugh Dance <hugh.dance.15@ucl.ac.uk>.

*Proceedings of the 42$^{nd}$ International Conference on Machine Learning*, Vancouver, Canada. PMLR 267, 2025. Copyright 2025 by the author(s).

## 2. Background and Problem Setup

In this section, we briefly review how MCMC algorithms are vectorized, explain the synchronization problems that can arise, and formalize the problem mathematically.

### 2.1. MCMC Algorithms and Vectorization

MCMC methods aim to draw samples from a target distribution $\pi$ (typically on a subset of $\mathbb{R}^d$) which is challenging to sample from directly. To do so, they generate samples $\{\boldsymbol{x}_1, ..., \boldsymbol{x}_n\} \in \mathbb{R}^{d \times n}$ from a Markov chain with transition kernel $P$ and invariant distribution $\pi$, by starting from an initial state $\boldsymbol{x}_0 \in \mathbb{R}^d$ and iteratively sampling $\boldsymbol{x}_{i+1} \sim P(\cdot|\boldsymbol{x}_i)$. For aperiodic and irreducible Markov chains, as $n \to \infty$ the samples will converge in distribution to $\pi$ (Brooks et al., 2011). In practice, $P$ is implemented by a deterministic function `sample`, which takes in the current state $\boldsymbol{x}_i$ and a pseudo-random state $r_i \in \mathbb{N}$, and returns new states $\boldsymbol{x}_{i+1}$ and $r_{i+1}$. This procedure is given in Algorithm 1.

---

**Algorithm 1** MCMC algorithm with `sample` function

1: **Inputs**: sample $\boldsymbol{x}_0$, seed $r_0$
2: **for** $i \in \{1, ..., n\}$ **do**
3:     generate $\boldsymbol{x}_i, r_i \leftarrow$ `sample`$(\boldsymbol{x}_{i-1}, r_{i-1})$
4: **end for**
5: **return** $\boldsymbol{x}_1, \ldots, \boldsymbol{x}_n$

---

Practitioners commonly run Algorithm 1 on $m$ different initializations, producing $m$ chains of samples. Given an implementation of `sample`: $\mathbb{R}^d \times \mathbb{N} \to \mathbb{R}^d \times \mathbb{N}$, one way to do this is to use an automatic vectorization tool like JAX's `vmap`[1]. `vmap` takes `sample` as an input and returns a new program, `vmap(sample)`: $\mathbb{R}^{d \times m} \times \mathbb{N}^m \to \mathbb{R}^{d \times m} \times \mathbb{N}^m$, which operates on a batch of inputs collected into tensors

$$\tilde{\boldsymbol{x}}_{i-1} := (\boldsymbol{x}_{i-1,1}, \ldots, \boldsymbol{x}_{i-1,m}) \in \mathbb{R}^{d \times m}$$
$$\tilde{r}_{i-1} := (r_{i-1,1}, \ldots, r_{i-1,m}) \in \mathbb{N}^m$$

and returns the corresponding outputs from `sample`. One can therefore turn Algorithm 1 into a multi-chain algorithm by simply replacing `sample` with `vmap(sample)` and replacing $(\boldsymbol{x}_0, r_0)$ by $(\tilde{\boldsymbol{x}}_0, \tilde{r}_0)$ (see Algorithm 6 in Appendix B). Under the hood, `vmap` transforms every instruction in `sample` (e.g., a dot product) into a corresponding instruction operating on a batch of inputs (e.g., matrix-vector multiplication); that is, it 'vectorizes' `sample`. These instructions are executed in lock-step across all chains. Using `vmap` usually yields code that performs as well as manually-batched code. For this reason, as well as its simplicity and composability with other transformations like `grad` (for automatic differentiation) and `jit` (for Just-In-Time compi-

---

[1] We use JAX and its vectorization map `vmap` throughout, since this framework is widely adopted. Similar constructs exist in TensorFlow (`vectorized_map`) and PyTorch (`vmap`).

---

lation), `vmap` has been adopted by major MCMC libraries such as NumPyro and BlackJAX.

### 2.2. Synchronization Problems with While Loops

Control flow (i.e., `if/else`, `while`, `for`, etc.) poses a challenge for vectorization, because different batch members may require a different sequence of instructions. `vmap` solves this by executing all instructions for all batch members, and masking out irrelevant computations. As a consequence, if `sample` contains a `while` loop, then `vmap(sample)` will execute the body of this loop for all chains until all termination conditions are met. Until then, no further instruction can be executed. As a result, if there is high variation in the number of loop iterations across chains, running Algorithm 1 with `vmap(sample)` introduces a synchronization barrier across all chains: *at every iteration, each chain has to wait for the slowest* `sample` *call*.

This issue arises in practice, because a number of important MCMC algorithms have while loops in their `sample` implementations: such as variants of slice sampling (Neal, 2003; Murray et al., 2010; Cabezas and Nemeth, 2023), delayed rejection methods (Mira et al., 2001; Modi et al., 2024), the No-U-Turn sampler (Hoffman et al., 2014), and unbiased Gibbs sampling (Qiu et al., 2019).

**Formalizing The Problem.** Here we formalize the problem through a series of short derivations. These will be made precise in Section 4. Suppose we run a `vmap`'ed version of Algorithm 1 using a `sample` function that has a while loop. Let $N_{i,j}$ denote the number of iterations required by the $j$th chain to obtain its $i$th sample. If the while loop has a variable length, $N_{i,j}$ is a random number. Due to the synchronization problem described above, the time taken to run `vmap(sample)` at iteration $i$ is approximately proportional to the largest $N_{i,j}$ out of the $m$ chains, $\max_{j \leq m} N_{i,j}$. If we ignore overheads, the total runtime after $n$ samples, $C_0(n)$, is then roughly proportional to

$$C_0(n) \propto \sum_{i=1}^{n} \max_{j \leq m} N_{i,j} \qquad (1)$$

By contrast, if the chains could be run without any synchronization barriers, the time taken would instead be

$$C_*(n) \propto \max_{j \leq m} \sum_{i=1}^{n} N_{i,j} \qquad (2)$$

The key difference is that the maximum is now outside the sum. This reflects the fact that when running independently, we only have to wait at the end for the slowest chain to collect its $n$ samples, rather than waiting at every iteration. Clearly $C_*(n) \leq C_0(n)$. Significant speed-ups are obtainable by de-synchronizing the chains when $C_*(n) \ll C_0(n)$. If each $N_{i,j}$ converges in distribution to some $\mathbb{P}_N$ as we

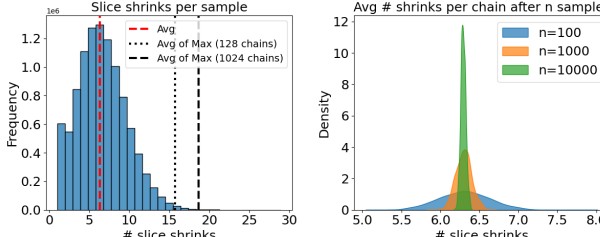

*Figure 1.* Statistics on the elliptical slice sampler for Gaussian Process Regression on the Real Estate Dataset (Yeh, 2018). LHS: histogram of the number of slice shrinks per sample. RHS: smoothed histogram of the average number of slice shrinks per sample across $m = 1024$ chains, after $n \in \{100, 1000, 10000\}$ samples.

draw more samples, and an appropriate central limit theorem holds (so that the sums in (1) and (2) behave like scaled means), we can expect for large enough $n$ that:

$$C_0(n) = \mathcal{O}_p(n \, \mathbb{E} \max_{j \leq m} N_{\infty,j}) \tag{3}$$

$$C_*(n) = \mathcal{O}_p(\max_{j \leq m} n \, \mathbb{E} N_{\infty,j}) = \mathcal{O}_p(n \, \mathbb{E} N_{\infty,1}) \tag{4}$$

where the last equality assumes $(N_{\infty,1}, ..., N_{\infty,m}) \overset{iid}{\sim} \mathbb{P}_N$. De-synchronizing the chains will result in large speedups if

$$\mathbb{E} \max_{j \leq m} N_{\infty,j} \gg \mathbb{E} N_{\infty,1} \tag{5}$$

We will see that Equation (5) holds in various situations.

**Example.** Consider the elliptical slice sampling algorithm (Murray et al., 2010) which samples from distributions which admit a density with Gaussian components, $p(\boldsymbol{x}) \propto f(\boldsymbol{x})\mathcal{N}(\boldsymbol{x}|0, \Sigma)$. Its transition kernel (see Algorithm 8 in Appendix B) draws each sample by (i) generating a random ellipse of permitted moves and an initial proposal, and (ii) iteratively shrinking the set of permitted moves and resampling the proposal from this set until it exceeds a log-likelihood threshold. The second stage uses a while loop which requires a random number of iterations.

On Figure 1 we display results when implementing this algorithm in JAX to sample from the hyperparameter posterior of a Gaussian process implemented on a regression task using a real dataset from the UCI repository (details are in Section 7). On the LHS we can see the distribution of the number of while loop iterations (i.e. slice shrinks) needed to generate a sample. While the average is ~6, the average of the *maximum* across 1024 chains is >18, which implies $\mathbb{E} \max_{j \leq 1024} N_{n,j} \approx 3\mathbb{E} N_{n,1}$. On the RHS, we can see that the differences across the chains do balance out as more samples are drawn, which suggests a certain central limit theorem may hold. In particular, after just 100 samples the distribution of the *average* number of iterations

per chain is contained in the interval $[5, 8]$, and after 10,000 samples this shrinks to $[6.2, 6.4]$. This implies that if we could vectorize the algorithm without incurring synchronization barriers, we could improve the Effective Sample Size per Second (ESS/Sec) by up to 3-fold. We will see that for other algorithms (e.g., HMC-NUTS and delayed rejection) the potential speed-ups are much larger than this.

## 3. Finite State Machines for MCMC

In this section we present an approach to implementing MCMC algorithms, which avoids the above synchronization problems when vectorizing with `vmap`. The basic idea is to break down the transition kernel (`sample`) into a series of smaller 'steps' which avoid iterative control flow like while loops and have minimal variance in execution time. We then define a runtime procedure that allows chains to progress through their own step sequences in de-synchronized fashion. To do this in a principled manner, we adapt the framework of finite state machines (FSMs) (Hopcroft et al., 2001). An FSM is a 5-tuple $(\mathcal{S}, \mathcal{Z}, \delta, \mathcal{B}, \mathcal{F})$, where $\mathcal{S}$ is a finite set of states, $\mathcal{Z}$ is a finite input set, $\delta : S \times \mathcal{Z} \to S$ is a transition function, $\mathcal{B} \in S$ is an initial state, and $\mathcal{F}$ is the set of final states. We deviate from the usual definition by allowing each state to be a function $S : \mathcal{Z} \to \mathcal{Z}$ that updates inputs dynamically[2]. Below we show how to represent different `sample` functions as an FSM.

### 3.1. `sample`-to-FSM conversion

At a high-level, we construct the FSM of an MCMC algorithm as follows. The states $\mathcal{S} = (S_1, ..., S_K)$ are chosen as functions $\mathcal{Z} \to \mathcal{Z}$ which execute contiguous code blocks of the algorithm. The boundaries of each code block are delineated by the start and end of any while loops. For example, $S_1$ executes all lines of code before the first while loop, $S_2$ executes all lines of code from the beginning of the first while loop body to either the start of the next while loop or the end of the current while loop, and so on. The inputs $z \in \mathcal{Z}$ are all variables used in each code block. (e.g. current sample $\boldsymbol{x}$, proposal $\boldsymbol{x}'$, log-likelihood $\log p(\boldsymbol{x})$[3], whilst the transition function $\delta$ selects the next code block to run according to the relevant loop condition and the received output $z'$ after executing the current block (e.g. if a while loop starts after the block $S_1$ executes, $\delta$ will check this loop's condition using the output $z' = S_1(z)$). Below we make this construction procedure more precise for three kinds of `sample` functions which cover all MCMC algorithms considered in the present work: (i) functions with a single while loop, (ii) functions with two sequential while

---

[2]This essentially results in a so-called 'Finite State Transducer' with a different runtime than typical.

[3]Typically $\boldsymbol{x} \in \mathbb{R}^d$ so $\mathcal{Z}$ is not finite. However, in practice all samples lie in a finite subset of $\mathbb{R}^d$ (e.g. 32-bit floating points).

loops, and (iii) functions with two nested while loops. An automated construction process for more general programs using a toy language is given in Appendix B.

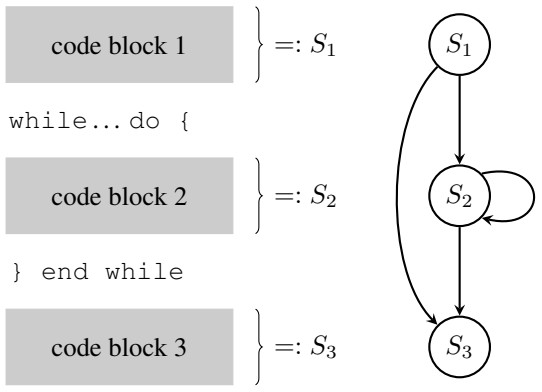

*Figure 2.* FSM of an MCMC algorithm with a single while loop.

**The Single While Loop Case.** The simplest case is a `sample` method with a single while loop. This covers (for example) elliptical slice sampling (Murray et al., 2010) and symmetric delayed rejection Metropolis-Hastings (Mira et al., 2001). In this case, we break `sample` down into three code blocks $B_1, B_2, B_3$ (one before the while loop, one for the body of the while loop, and one after the while loop) and the termination condition of the loop. This is shown on the LHS of Figure 2. Using these blocks, we define the FSM as $(\mathcal{S}, \mathcal{Z}, \delta, \mathcal{B}, \mathcal{F})$, where (1) $\mathcal{Z}$ is the set of possible values for the local variables of `sample` (e.g., the current sample $\boldsymbol{x}$, seed $r$, and proposal etc.), (2) $\mathcal{S} = \{S_1, S_2, S_3\}$ is a set of three functions $\mathcal{Z} \to \mathcal{Z}$, where for each $k \in \{1, 2, 3\}$, $S_k(z)$ runs $B_k$ on local variables $z$ and returns their updated value, (3) for each $k \in \{0, 1\}$ and $z \in \mathcal{Z}$, the transition function $\delta(S_k, z)$ checks the while loop termination condition using $z$, and returns $S_2$ if False and $S_3$ if True, (4) $\mathcal{B} = S_1$ and, finally, (5) $\mathcal{F} = S_3$. The RHS of Figure 2 illustrates the resulting FSM diagram. Note there is an edge $S_k \to S_{\tilde{k}}$ between states if and only if $\delta(S_k, z) = S_{\tilde{k}}$ for some $z$.

**Two Sequential While Loops.** We break down `sample` into two blocks: $B_1$ contains all the code up to the second while loop. $B_2$ contains the remaining code. In this case, $B_1$ and $B_2$ are now single while loop programs, and thus can both be represented by FSMs $F_1$ and $F_2$ using the above rule. The FSM representation of `sample` can then be obtained by "stitching together" $F_1$ and $F_2$. The full construction process is in Appendix B. The resulting FSM is provided for the case of the Slice Sampler (Figure 3, top-right panel), which contains two[4] while loops: one for expanding the slice, and one for contracting the slice.

**Two Nested While Loops.** In the case of two nested while

---

[4]For 1D problems, the slice expansion loop can be broken into two loops (for the upper and lower bound of the interval).

loops, we break down `sample` into $B_1, B_2, B_3$, where $B_1$ (resp. $B_3$) is the code before (resp. after) the outer while loop and $B_2$ is the outer while loop body. As $B_2$ is a single-while loop program, it admits its own FSM $F_i$. Building the final FSM of `sample` then informally consists in obtaining a first, "coarse" FSM by treating $B_2$ as opaque, and then refining it by replacing $B_2$ with its own FSM. The full construction process is given in Appendix B. The resulting FSM is provided for the case of NUTS (Figure 3, bottom-right panel), which—in its iterative form—uses the outer while loop to determine whether to keep expanding a Hamiltonian trajectory, and the inner while loop to determine whether to keep integrating along the current trajectory.

### 3.2. Defining the FSM Runtime

Going forward, for convenience we assume the transition function $\delta$ takes in and returns the *label* $k$ of each block $S_k$, rather than $S_k$ itself. Now, given a constructed FSM, in Algorithm 2 we define a function `step`, which when executed performs a single transition along an edge in the FSM graph. For reasons made clear shortly, we augment `step` to return a flag indicating when the final block is run.

---

**Algorithm 2** `step` function for FSM

**Inputs:** algorithm state $k$, variables $z$
1: set `isSample` $\leftarrow 1\{k = K\}$
2: $z \leftarrow$ `switch`$(k, [\{\text{run } S_1(z)\}, ..., \{\text{run } S_K(z)\}])$
3: $k \leftarrow$ `switch`$(k, [\{\text{run } \delta(1, z)\}, ..., \{\text{run } \delta(K, z)\}])$
4: **return** $(k, z, \text{isSample})$

---

Here `switch` uses the value of $k$ to determine which branch to run. If we start from some input $(k, z) = (0, z_0)$ and call `step` repeatedly, we transition through a sequence of blocks of `sample`, until we eventually reach the terminal state. At this point, a sample is obtained (as indicated by `isSample=True`). We can use this function to draw $n$ samples, by (i) adding a transition from the terminal state $\mathcal{F} = S_K$ back to the initial state $\mathcal{B} = S_0$, and (ii) defining a wrapper function which iteratively calls `step` until `isSample=True` is obtained $n$-times (see Algorithm 3).

---

**Algorithm 3** FSM MCMC algorithm

1: **input:** initial value $\boldsymbol{x}_0$, # samples $n$
2: **initialize:** $z = \text{init}(\boldsymbol{x}_0)$, $X = \text{list}()$, $B = \text{list}()$
3: Set $t = 0$ and $k = 0$
4: **while** $t < n$ **do**
5:     $(k, z, \text{isSample}) \leftarrow$ `step`$(k, z)$
6:     append current sample value $\boldsymbol{x}$ stored in $z$ to $X$
7:     append `isSample` to $B$
8:     update sample counter $t \leftarrow t + \text{isSample}$
9: **end while**
10: **return** $X[B]$

---

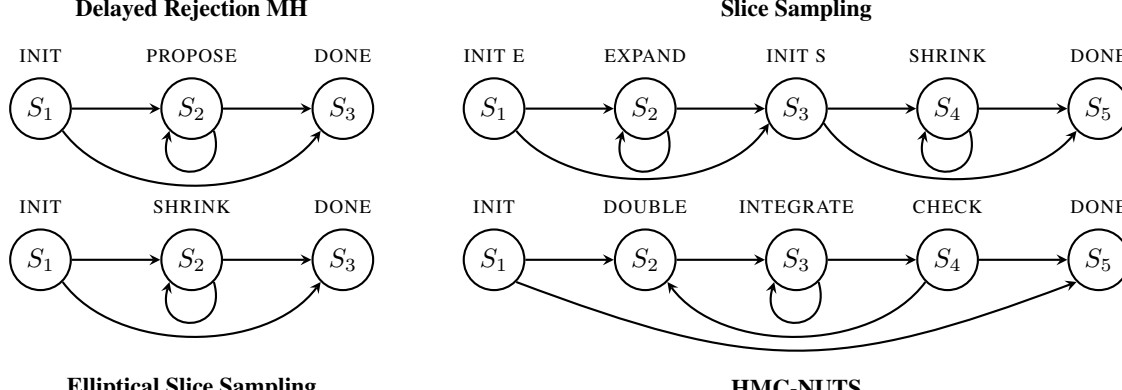

*Figure 3.* Finite state machines for the `sample` function of different MCMC algorithms: The symmetric delayed-rejection Metropolis-Hastings algorithm (Mira et al., 2001), elliptical slice sampling (Murray et al., 2010), (vanilla) slice sampling (with single slice expansion loop) (Neal, 2003), the No-U-Turn sampler for Hamiltonian Monte Carlo (Hoffman et al., 2014).

Both Algorithm 3 and Algorithm 1 call `sample` $n$-times and can be easily vectorized with `vmap`. For Algorithm 3, we just call `vmap` on `step` and modify the outer loop to terminate when all chains have collected $n$ samples each (see Algorithm 7 in Appendix B). The crucial difference is: (i) by defining a 'step' to be agnostic to which block each chain is currently on, Algorithm 3 enables chains to simultaneously progress their independent block sequences, and (ii) by moving all while loops to the outer layer, Algorithm 3 only requires chains to synchronize after $n$ samples.

## 4. Time Complexity Analysis of FSM-MCMC

One limitation with our FSM design is the `step` function relies on a switch to determine which block to run. When vectorized with `vmap` or an equivalent transformation, all branches in `switch` are evaluated for all chains, with irrelevant results discarded. This means the cost of a single call of `step` is the cost of running all state functions $S_1, ..., S_K$. To obtain a speedup, the FSM must therefore sufficiently decrease the expected number of steps to obtain $n$ samples from each chain. In this section we quantitatively derive conditions under which this occurs in the simplified setting of an MCMC algorithm with a single while loop. This enables us to subsequently optimize the FSM design in Section 5.

### 4.1. Long Run Costs of FSM and Non-FSM MCMC

To this end, consider a `sample` function with a single while loop. Suppose it is broken down into $K$ blocks $\{S_1, ...S_K\}$, by our FSM construction procedure, where $S_k$ (for some $k \in [K]$) executes the body of the loop. Note our procedure yields $K \leq 3$ for one while loop, but we relax this here to analyze the effect of the number of blocks on performance. A single call of `vmap(sample)` executes $S_l$ once if $l \neq k$, and $\max_j N_j$-times if $l = k$, where $N_j = $ # loop iterations

for chain $j$. We assume the cost of executing each block $S_l$ with `vmap` is $c_l(m)$ for $m$ chains (here the dependence on $m$ reflects underlying GPU vectorization efficiency). The average cost per sample of Algorithm 1 after $n$ samples, when vectorized for $m$ chains, is then

$$C_0(m,n) := A_0(m) + B_0(m)\frac{1}{n}\sum_{i=1}^{n}\max_{j\in[m]} N_{i,j} \quad (6)$$

where $A_0(m) := \sum_{j\neq k} c_j(m)$, $B_0(m) = c_k(m)$, and $N_{i,j}$ = # calls of the loop body $S_k$ for chain $j$ and sample $i$. The max reflects the synchronization barrier across chains.

In this case, for the FSM (`vmap`'ed Algorithm 3), the cost of calling each block $S_l$ is $\sum_{j=1}^{K} c_j(m)$, since `vmap(step)` executes all `switch` branches for all chains. We will assume this cost can be scaled by some $\alpha \in [\max_{k\in[K]} c_k(m)/\sum_{k=1}^{K} c_k(m), 1]$, since we will later redesign `step` to reduce $\alpha < 1$. This gives an average cost per sample for $m$ chains after $n$ samples of,

$$C_F(m,n) = A_F(m) + B_F(m)\max_{j\in[m]}\frac{1}{n}\sum_{i=1}^{n} N_{i,j} \quad (7)$$

where $A_F(m) = \alpha(K-1)(c_{\neg k}(m) + c_k(m))$, $B_F(m) = \alpha(c_k(m) + c_{\neg k}(m))$, and $c_{\neg k}(m) := \sum_{j\neq k} c_j(m)$.

Let $\boldsymbol{X}_{i,j}$ be the $i^{th}$ random sample obtained by chain $j$. If the joint sequence $(\boldsymbol{X}_{i,j}, N_{i,j})_{i\geq 1}$ is an appropriate Markov chain, we have the following concentration result for $C_0, C_F$, which formalizes our derivations in Section 2.2.

**Theorem 4.1** (Long Run Costs). *Let $N_{i,j} \in [0, B]$, $\boldsymbol{X}_{i,j} \in \mathcal{X} \subseteq \mathbb{R}^d$, $(\boldsymbol{X}_{i,j}, N_{i,j})_{i\geq 1}$ be a Markov Chain with stationary distribution $\pi$ with absolute spectral gap $1 - \lambda \in [0, 1)$. Then with probability $1 - \delta$ we have the inequalities:*

$$|C_0(m,n) - C_0(m)| \leq MB_0(m)n^{-\frac{1}{2}}\ln(2/\delta) \quad (8)$$

$$|C_F(m,n) - C_F(m)| \leq MB_F(m)n^{-\frac{1}{2}}\ln(2m/\delta) \quad (9)$$

*Where, for some* $(N_1, .., N_m) \overset{iid}{\sim} \pi_N$, *the long run costs are*

$$C_0(m) := A_0(m) + B_0(m)\mathbb{E}_\pi \max_{j \in [m]} N_j \qquad (10)$$

$$C_F(m) := A_F(m) + B_F(m)\mathbb{E}_\pi N_1 \qquad (11)$$

The result says that as $n \to \infty$, the cost per sample of the standard MCMC design converges to the expected time for the *slowest* chain to draw a sample, whilst the FSM design converges to the expected time for a *single* chain to draw a sample, scaled by the additional cost of control-flow in `step` (i.e. $A_F(m)$ and $B_F(m)$). Both rates are $\mathcal{O}(n^{-\frac{1}{2}})$.

### 4.2. The Relative Long-Run Cost of the FSM

Using the form of the constants in Theorem 4.1, the ratio of the long-run costs, $E(m) := C_0(m)/C_F(m)$, is given by

$$E(m) = \frac{c_{\neg k}(m) + c_k(m)\mathbb{E}\max_{j \in [m]} N_j}{\alpha(c_{\neg k}(m) + c_k(m))(K - 1 + \mathbb{E}N_1)} \qquad (12)$$

In Proposition A.3 in Appendix A we prove that:

$$E(m) \leq R(m) := \frac{\mathbb{E}\max_{j \in [m]} N_j}{\mathbb{E}N_1} \qquad (13)$$

and that this bound is tight. We refer to $R(m)$ as the 'theoretical efficiency bound' for the FSM. Note from Equation (12) that minimizing $\alpha$ and $K$ improves the efficiency $E(m)$ of the FSM. In Section 5 we introduce two techniques to minimize $\alpha$ and $K$ in practice, which enable us to nearly obtain the efficiency bound $R(m)$ for certain MCMC algorithms.

**Scale of Potential Speed-ups.** The size of $R(m)$ depends (only) on the underlying distribution of $N_1$ (since $N_i =_d N_j$ $\forall i, j \in [m]$). Whenever $N_1$ is sub-exponential, it is known that $R(m) = \mathcal{O}(\ln(m))$ (Vershynin, 2018). Although this implies a slow rate of increase in $m$, $R(m)$ can be still be very large for small values of $m$. For example, if $N_j/B \sim Bern(p)$ (i.e., one either needs zero or $B$ iterations to get a sample), then $R(m) = (1 - (1 - p)^m)/p$ and converges to $1/p$ exponentially fast as $m$ increases. For small $p$ this can be very large even for small $m$. In general $R(m)$ is sensitive to the skewness of the distribution: distributions on $[0, B]$ with zero skewness have $R(m) = 2$, whilst distributions with skew of 10 can have $R(m) \approx 100$; see Figure 4. Intuitively, these are the distributions where chains are slow only occasionally, but at least one chain is slow often. In such cases, our FSM-design can lead to enormous efficiency gains, as we show in experiments.

## 5. Optimal FSM design for MCMC algorithms

In this section we provide two strategies to modify the function `step` to (effectively) reduce $\alpha$ and $K$. These strategies enable us to develop MCMC implementations which nearly obtain the theoretical bound $R(m)$ in some experiments.

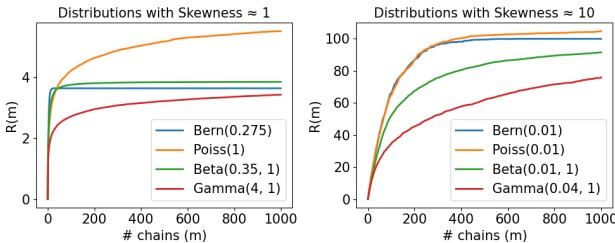

*Figure 4.* $R(m) = \frac{\mathbb{E}\max_{j \in [m]} N_j}{\mathbb{E}M_1}$ for different distributions with skewness $\gamma_1 \approx 1$ (LHS) and $\gamma_1 \approx 10$ (RHS).

### 5.1. Step Bundling to Reduce $K$

Given an FSM with `step` defined by code blocks $S_1, ..., S_K$ and transition function $\delta$, one can 'bundle' multiple steps together using a modified step function, `bundled_step`, which replaces the switch over the algorithm state $k$ in `step` with a series of separate conditional statements. This is shown in Algorithm 4 for an example with two state functions. Note that under the "run all branches and mask" behaviour of `vmap`, `vmap(step)` and `vmap(bundled_step)` have the same cost. However, whenever `bundled_step` runs $S_1$ and $\delta$ returns $k = 2$, it immediately also runs $S_2$. This reduces the 'effective' number of states $K$ and/or the overall number of steps needed to recover a sample, increasing efficiency. In principle, the block ordering can be optimized for sequences that are expected to occur with higher probability. However, we found chronological order a surprisingly effective heuristic.

---

**Algorithm 4** `bundled_step` for FSM with $S_1, S_2$

---

**Inputs:** algorithm state $k$, variables $z$

1: set `isSample` $\leftarrow 1\{k = 2\}$
2: **if** $k = 1$ **then**
3:     run block $S_1$ with local variables $z$
4:     update state $k \leftarrow \delta(1, z)$
5: **end if**
6: **if** $k = 2$ **then**
7:     run block $S_2$ with local variables $z$
8:     update state $k \leftarrow \delta(2, z)$
9: **end if**
10: **return** $(k, z, \text{isSample})$

---

### 5.2. Cost Amortization to Reduce $\alpha$

If a function $g$ is called on a variable $\theta \in z$ inside $M \leq K$ state functions, a single call of `vmap(step)` (or `vmap(bundled_step)`) will execute $g$ $M$-times. To prevent this, we (i) augment `step` to return a flag `doComputation` indicating if $g$ executed in the next code block, and (ii) define a new step function `amortized_step` around `step` which calls `step` once, and executes $g$ if `doComputation=True`. The resulting step function is shown in Algorithm 5 (note if $g$ is called in step $S_1$, we assume it is already pre-computed in $z$).

Note the function $g$ is now only called once per step when using `vmap` on `amortized_step`. In particular, if the executions of $g$ cost (in total) $\beta \in [0, 1]$ fraction of the total cost of all blocks, amortization will reduce this fraction to $\frac{\beta}{M} + (1 - \beta)$ We indeed observe this when amortizing $g = \log p$ for the elliptical slice sampler in Section 7.1. In that scenario, $M = 2$ and $\beta \approx 1$ as the log-likelihood is very expensive, resulting in a $2\times$ speed-up over no amortization. One issue is that amortization only allows steps to be bundled which do not require calls of $g$. We adapt the algorithm to handle this in Appendix B.

---

**Algorithm 5** `amortized_step` for FSM with function $g$

---

1: **Input:** Algorithm state $k$, variables $z$
2: $(k, z, \texttt{isSample}, \texttt{doComputation}) \leftarrow \texttt{step}(k, z)$
3: **if** doComputation **then**
4:     Unpack state $(z', \theta) = z$
5:     Do computation $\theta \leftarrow g(z)$
6:     Re-pack state $z \leftarrow (z', \theta)$
7: **end if**
8: **Return** $(k, z, \texttt{isSample})$

---

## 6. Related work

**FSMs in Machine Learning.** Previous work in machine learning has used the framework of FSMs to design image-based neural networks (Ardakani et al., 2020) and Bayesian nonparametric time series models (Ruiz et al., 2018), as well as extract representations from Recurrent-Neural-Networks (RNNs) (Muškardin et al., 2022; Cechin et al., 2003; Tiňo et al., 1998; Zeng et al., 1993; Koul et al., 2018; Svete and Cotterell, 2023). To our knowledge, our work is the first that uses FSMs as a framework to represent MCMC algorithms, and design novel implementations.

**Efficient MCMC on Modern Hardware.** Given the recursive nature of HMC-NUTS, previous work has reformulated the algorithm for compatibility with machine learning frameworks that cannot naively support recursion (Abadi et al., 2016; Phan et al., 2019; Lao et al., 2020). However, these implementations do not address synchronization inefficiencies caused by automatic vectorization tools. Our work bears some similarities with a general-purpose algorithm proposed in the High-Performance-Computing literature for executing batched recursive programs (Radul et al., 2020). Both their method and ours breaks programs down into smaller blocks for efficient vectorization, but there are several major differences. Our approach provides a recipe for implementing single-chain iterative MCMC algorithms and is fully compatible with automatic vectorization tools like `vmap`. In contrast, their algorithm is designed for recursive programs and requires code which is already batched. Crucially, to avoid synchronization barriers due to while loops, this code cannot have been batched with a `vmap`-equivalent vectorization tool, since `vmap` converts while loops into

a single batched primitive that cannot be broken down by their algorithm. Additionally, our method uses the fewest (while-loop-free) blocks possible and allows for control flow within blocks (which we showed is crucial for optimal performance under the "run and mask" paradigm). By contrast, their algorithm does not allow blocks to contain any control-flow, yielding many small blocks. We also obtain provable speed-ups for MCMC via theoretical analysis.

## 7. Experiments

The following experiments evaluate FSMs for different MCMC algorithms with while loops against their standard (non-FSM) implementations, as well as other MCMC methods in one experiment. All methods consist of single chain MCMC algorithms written in JAX, turned into multi-chain methods with `vmap`, and compiled using `jit`. All experiments are run in JAX on an NVIDIA A100 GPU with 32GB CPU memory. Code can be found at https://github.com/hwdance/jax-fsm-mcmc.

### 7.1. Delayed-Rejection MH on a Simple Gaussian

We first illustrate basic properties of the FSM conversion on a toy example. The MCMC algorithm used is symmetric Delayed-Rejection Metropolis Hastings (DRMH) (Mira et al., 2001), which is a simple example of a delayed rejection method. Symmetric DRMH modifies the Random-Walk Metropolis-Hastings algorithm by iteratively re-centering the proposal distribution on the rejected sample and resampling until either acceptance occurs or a maximum number of tries $M$ is reached. To ensure detailed balance, the acceptance probability is adjusted to account for past rejections.

**Experimental setup**. We implement symmetric DRMH using (`vmap`'ed) Algorithm 1 (baseline) and Algorithm 3 (ours) to sample from a univariate Gaussian $\mathcal{N}(0, 1)$, varying the number of chains. We use a $\mathcal{N}(x, 0.1)$ proposal distribution with $M = 100$ tries per sample and draw 10,000 samples per chain. Although DRMH has 3 state functions by default (see Figure 3), the INIT and DONE states are essentially empty, and so just a single state function can be used for the while loop body. This means an appropriate FSM implementation should be able to get close to $R(m)$, up to overheads. To illustrate the importance of designing FSMs with as few (effective) state functions as possible, we implement an FSM which unrolls the while loop body into four different state functions, as well as a (condensed) FSM with a single state function (i.e. effectively uses bundling).

**Results**. As the number of chains $m$ increases, the FSM implementations increasingly outperform the standard implementation (see Figure 6). This reflects the increasing synchronization cost of waiting for the slowest chain. The speedups are near an order of magnitude when $m = 1024$

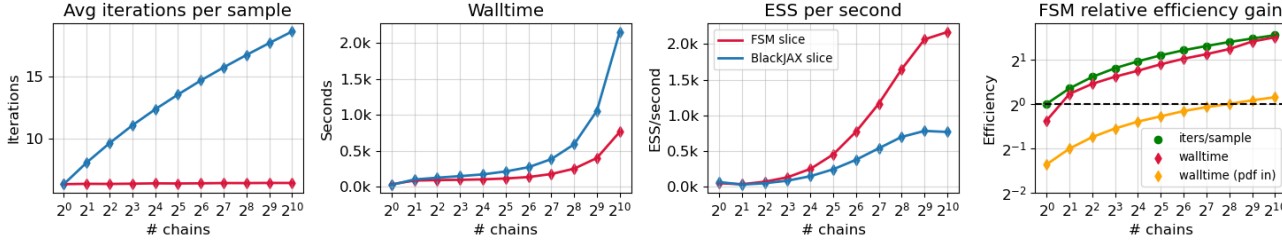

*Figure 5.* Average results using 10 random seeds (standard deviations too small to show) from drawing 10k posterior samples for the covariance hyperparameters $(\tau, \theta, \sigma)$ of a Gaussian Process $Y(x) = f(x) + \epsilon$ on the Real Estate UCI dataset ($n = 411, d = 6$). Blue = BlackJAX elliptical slice, Red = FSM elliptical slice sampler (red). LHS: the average number of sub-iterations (i.e., ellipse contractions) needed to draw a single sample increases from 6 (1 chain) to 18 (1024 chains) for the standard implementation due to synchronization barriers, but remains constant for our FSM. Middle two plots: The FSM can run $\sim$3x faster by avoiding synchronization barriers, as shown by the Walltime (left-middle) and ESS/S (right-middle). RHS: the ratio of average iterations per sample (i.e. $R(m)$) (green) bounds the obtainable 'efficiency gain' using our FSM, but is nearly obtained in relative walltime when amortizing log-pdf calls (red).

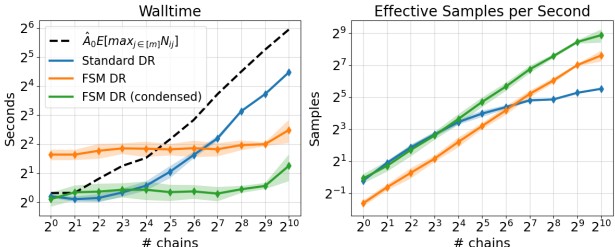

*Figure 6.* Mean and standard deviation walltimes (LHS) and ESS (RHS) of the symmetric Delayed-Rejection Algorithm (Mira et al., 2001) using Algorithm 1 and our FSM implementation Algorithm 3 (both with vmap), on a univariate Gaussian (10 random seeds). FSM uses step with the while loop body (PROPOSE in Figure 3) split into 4 states, whilst FSM-condensed uses a single step. Using a single step leads to a $\sim$ 3x efficiency gain.

for the condensed FSM. Bundling sees nearly a 3x efficiency gain and enables no performance loss when $m = 1$ and there is no synchronization barrier. Note Standard DR tracks the profile of the estimated $\mathbb{E}[\max_{j \in [m]} N_{1,j}]$, whilst the FSMs track the profile of $\mathbb{E}[N_{1,j}]$ (which is flat). This implies the condensed FSM (i.e. with step bundling) has been able to approximately obtain $R(m)$ up to a constant factor.

### 7.2. Elliptical Slice Sampling on Real Estate Data

The elliptical slice sampler (introduced in the example in Section 2.2) has a single while loop, resulting in three state functions (see Figure 3). We compare BlackJAX's implementation to our FSM implementation.

**Experimental setup**. We apply the sampler to infer posteriors on covariance hyperparameters in Gaussian Process Regression, using the UCI repository Real Estate Valuation dataset (Yeh, 2018). This dataset is comprised of $n = 414$ input and output pairs $\mathcal{D}_n = \{\boldsymbol{x}_i, y_i\}_{i=1}^n$, where $y_i$ is the house price of area $i$, and $\boldsymbol{x}_i \in \mathbb{R}^6$ are house price predictors including house age, spatial co-ordinates, and number of nearby convenience stores. We model $y = f(x) + \epsilon$

and assume $\epsilon \sim \mathcal{N}(0, \sigma^2)$, $f \sim \mathcal{GP}(0, k)$ with kernel $k(x, x') = \tau^2 \exp(-\lambda^2 |x - x'|^2)$. We use Normal priors $\sigma, \tau, \lambda \sim \mathcal{N}(0, 1)$, (so the ellipse is drawn using $\mathcal{N}(0, I)$) and use the sampler to draw $10k$ samples per chain from the posterior $p(\sigma, \tau, \lambda | \mathcal{D}_n)$, for varying # chains $m$.

**Results** are shown in Figure 5. As expected, the BlackJAX implementation suffers from synchronization barriers at every iteration due to using vmap with while loops: its average number of iterations per sample increases roughly logarithmically from 6 (1 chain) to 18 (1024 chains), whereas the FSM implementation remains constant. As a result, the FSM significantly improves walltime and ESS/second performance (Figure 5/middle). For instance, when 1024 chains are used, the FSM reduces the time to draw 10k samples per chain from over half an hour to about 10 minutes. The efficiency gain can be measured by the ratio (BlackJAX/FSM) of wall-times (shown in Figure 5/right). As expected, FSM efficiency increases with the number of chains. The greatest efficiency gain occurs precisely where the best ESS/second can be obtained via GPU parallelism. The analysis in Section 4 shows that the efficiency gain is upper-bounded by the ratio of average # iterations per sample for both methods (i.e., $R(m)$). This bound is almost achieved here by amortizing log-pdf calls. This is because (i) roughly 80% of the time is spent in the iterative 'SHRINK' state, and (ii) log-pdf calls dominate computational cost here, so amortizing them ensures the state function cost is similar to the standard implementation. Since the log-pdf is needed in two states, not amortizing it results in two log-pdf calls per step, and so we lose roughly a factor 2 in relative performance (orange line in Figure 5). These results change with data set size, which determines the cost of log-pdf calls (see Appendix B).

### 7.3. HMC-NUTS on High-Dimensional Correlated MoN

The NUTS variant of Hamiltonian Monte Carlo (Hoffman et al., 2014) adaptively chooses how many steps of Hamiltonian dynamics to simulate when drawing a sample, by

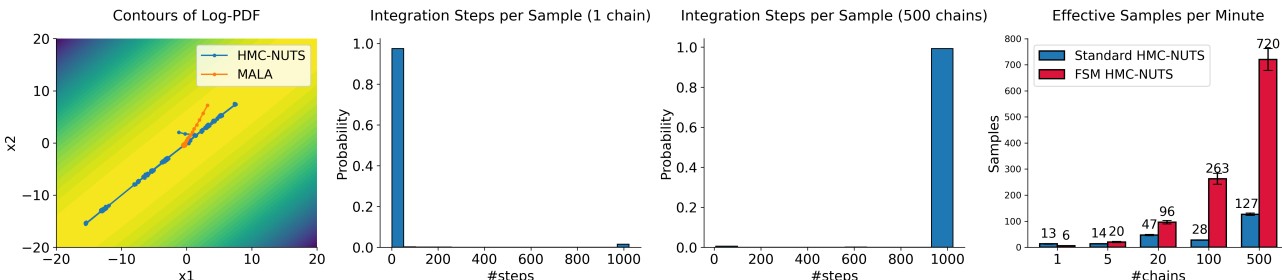

*Figure 7.* Left: contours of the first two dimensions of a correlated mixture of Gaussians, along with a single chain of HMC-NUTS and MALA. Middle Left: Histogram of the number of integration steps taken per sample for a single NUTS chain. Middle Right: histogram of the maximum number of integration steps taken per sample across 500 chains. Right: Effective samples per minute for the standard BlackJAX HMC-NUTS implementation, and our FSM implementation of HMC-NUTS (average and standard error bars from 5 seeds). The FSM achieves speed ups of nearly an order of magnitude for 100 chains, and more than half an order of magnitude for 500 chains.

checking whether the trajectory has turned back on itself or has diverged due to numerical error. Its iterative implementation in BlackJAX involves two nested while loops: An outer loop that expands the proposal trajectory, and an inner loop that monitors for U-turns and divergence. Converting these while loops into an FSM using our procedure results in five states (Figure 3). We again compare BlackJAX's implementation to our own (using `vmap` for both methods).

**Experimental setup**. We implement NUTS on a 100-dimensional correlated mixture of Gaussians ($\rho = 0.99$), with the mixture modes placed along the principal direction at $(-10 \cdot \mathbf{1}, \mathbf{0}, 10 \cdot \mathbf{1})$. We use a pre-tuned step-size with acceptance rate $\sim 0.85$ and identity mass matrix. We draw $n = 1000$ samples per chain and vary # chains $m$.

**Results**. The trajectory of a single NUTS chain (1000 samples) are displayed on the LHS of Figure 7 (one dot = one sample). The typical distance traveled by NUTS is small (few integration steps), with the occasional large jump (many integration steps) when the momentum sample aligns with the principal direction. In particular, the probability a sample requires less than 20 integration steps is $\sim 0.95$ and needs $> 1000$ steps is $\sim 0.01$, but the probability that *at least one* chain needs more than 1000 steps is $\sim 0.99$ (Middle Figure 7). This results in FSM speed-ups of nearly an order of magnitude for $m = 100$, and about half an order of magnitude for $m = 500$ (RHS Figure 7). Note that one can avoid sychronization barriers and obtain very high ESS/Sec using a simpler algorithm like MALA, but this fails to explore the distribution (LHS Figure 7).

### 7.4. Transport Elliptical Slice Sampling and NUTS on Distributions with Challenging Local Geometry

*Transport elliptical slice sampling* (TESS), due to Cabezas and Nemeth (2023), is a state-of-the-art variant of elliptical slice sampling designed for challenging local geometries. It uses a normalizing flow $T$ to 'precondition' the distribution $\pi$ into an approximate Gaussian, does elliptical slice sam-

*Table 1.* Left: ESS/sec for baselines on Predator Prey (PP), Google Stock (GS), German Credit (GC), Biochemical Oxygen Demand (BOD) distributions. Right: FSM speed-ups for NUTS/TESS.

| Dist. | ESS/sec (non-FSM) | | | | FSM Speed-up | |
|-------|--------|--------|--------|--------|--------|--------|
| | MEADS | CHEES | NUTS | TESS | NUTS | TESS |
| PP | 1.5 | nan | 0.02 | 2.3 | 1.5× | 1.7× |
| GS | 480.8 | 60.2 | 0.15 | 1116 | 3.5× | 2.2× |
| GC | 141.5 | 199.0 | 1.71 | 58.95 | 1.2× | 1.0× |
| BOD | 64.85 | 247.0 | 0.56 | 2978 | 0.8× | 1.1× |

pling on the transformed distribution $T_{\#}\pi$, and then pushes generated samples through $T^{-1}$ to recover samples from $\pi$. TESS achieves particularly good results on distributions with 'funnel' geometries, on which gradient-based methods like HMC-NUTS tend to struggle (Gorinova et al., 2020). TESS is similar to the elliptical slice sampler and so the FSM has the same 'single loop' structure (see Figure 3).

**Experimental setup** We examine the speed-ups using FSMs for TESS and NUTS on the four benchmark distributions in Cabezas and Nemeth (2023), which are chosen for their challenging geometries. As baselines we use two state-of-the-art adaptive HMC variants (MEADS (Hoffman and Sountsov, 2022), and CHEES (Hoffman et al., 2021)). We expect NUTS to perform poorly on these problems (as observed in Cabezas and Nemeth (2023)), but implementing it lets us at least examine the FSM speed-ups. For all methods, we average results over 128 chains of 1000 samples, with hyperparameters pre-tuned over 400 warm-up steps.

**Results** are in Table 1. TESS-FSM improved ESS/sec over TESS in all cases (in 2/4 cases by $\sim 2\times$) and in 3/4 cases improved the best ESS/sec. NUTS performed poorly as expected, but the FSM improved ESS/sec in 3/4 cases, and $> 3\times$ in one case. Speed-ups for NUTS and TESS were larger for tasks with an expensive log-pdf (PP, GS). This is because most time is spent in the loop states where the log-pdf is called, and a more costly log-pdf reduces the gap between the cost of executing these states for non-FSMs ($c_k$) and FSMs ($\sum_{j=1}^{m} c_j$) (see Section 4), improving efficiency.

## Acknowledgements

HD, PG and PO are supported by the Gatsby Charitable Foundation. This work was partially supported by NSF OAC 2118201.

## Impact Statement

This work improves the execution efficiency of MCMC algorithms. Our method has the potential to significantly reduce time, cost, and energy expenditure for a range of scientific applications.

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

# A. Mathematical Appendix

### A.1. Proof of Theorem 4.1

*Proof.* The result broadly follows from known Hoeffding bounds for Markov chains. For clarity we restate the relevant result below[5], using our notation and set-up.

We first define formally the notion of an absolute spectral gap, from (Fan et al., 2021).

**Definition A.1** (Absolute Spectral Gap). Let $\mathcal{Z}$ and $\pi$ denote the state space and the invariant measure of a Markov chain $\{Z_i\}_{i \geq 1}$. For a function $f : \mathcal{Z} \to \mathbb{R}$, write $\pi(h) := \int h(z)\,\pi(dz)$. Let $L_2(\pi) = \{h : \pi(h^2) < \infty\}$ be the Hilbert space consisting of $\pi$-square-integrable functions, and $L_2^0(\pi) = \{h \in L_2(\pi) : \pi(h) = 0\}$ be its subspace of $\pi$-meanzero functions. The transition probability kernel of the Markov chain, denoted by $P$, is viewed as an operator acting on $L_2(\pi)$. Let $\lambda \in [0, 1]$ be the operator norm of $P$ acting on $L_2^0(\pi)$. Then, $1 - \lambda$ is the *absolute spectral gap* of the Markov chain.

**Proposition A.2** (Hoeffding Bound for Markov Chains - Theorem 1 in Fan et al. (2021)). *Let $(Z_i)_{i \geq 1}$ be a Markov Chain with measurable state space $\mathcal{Z}$, stationary distribution $\pi$, and absolute spectral gap $1 - \lambda \in (0, 1]$. Let $f : \mathcal{Z} \to [0, B]$ be measurable and bounded. Then, for any $\epsilon > 0$,*

$$\mathbb{P}_\pi \left( \left| \sum_{i=1}^n f(Z_i) - n\mathbb{E}_\pi f(Z_1) \right| \geq \epsilon \right) \leq 2\exp\left( -\frac{1-\lambda}{1+\lambda} \frac{2\epsilon^2}{B^2} \right) \tag{14}$$

*where $1 - \lambda$ is the absolute spectral gap of $\pi$.*

Now we are ready to prove our results. We start with $C_0(m, n)$. First, note that since $(\boldsymbol{X}_{i,j}, N_{i,j})_{i \geq 1}$ is a Markov chain with stationary distribution $\pi$ for every chain $j$, then $(\{\boldsymbol{X}_{i,j}, N_{i,j}\}_{j=1}^m)_{i \geq 1}$ is also a Markov chain with stationary distribution $\pi^m := \pi \otimes \pi \underbrace{\otimes ... \otimes}_{m-1} \pi$, because the chains are independent. Therefore, if we set $Z_i := \{\boldsymbol{X}_{i,j}, N_{i,j}\}_{j=1}^m$ and $f(Z_i) := \max_{j \in [m]}(N_{i,j}) \in [0, B]$, we get by an application of Proposition A.2 that

$$\mathbb{P}_\pi \left( \left| \sum_{i=1}^n \max_{j \in [m]}(N_{i,j}) - n\mathbb{E}_\pi \max_{j \in [m]}(N_{1,j}) \right| \geq \epsilon \right) \leq 2\exp\left( -\frac{1-\lambda}{1+\lambda} \frac{2\epsilon^2}{nB^2} \right) \tag{15}$$

Setting $\delta = 2\exp\left( -\frac{1-\lambda}{1+\lambda} \frac{2\epsilon^2}{nB^2} \right)$ and re-arranging, we get that with probability $1 - \delta$ (under the stationary distribution $\pi$),

$$\left| \sum_{i=1}^n \max_{j \in [m]}(N_{i,j}) - n\mathbb{E}_\pi \max_{j \in [m]}(N_{1,j}) \right| \leq B\sqrt{\frac{n(1+\lambda)}{2(1-\lambda)} \ln\left(\frac{2}{\delta}\right)} \tag{16}$$

Multiplying by $B_0(m)$ and dividing by $n$ on both sides, and adding and subtracting $A_0(m)$ from the LHS gives us the result for $C_0$

$$\left| B_0(m)\frac{1}{n}\sum_{i=1}^n \max_{j \in [m]}(N_{i,j}) - B_0(m)\mathbb{E}_\pi \max_{j \in [m]}(N_{1,j}) \right| \leq B_0(m)B\sqrt{\frac{(1+\lambda)}{2n(1-\lambda)} \ln\left(\frac{2}{\delta}\right)} \tag{17}$$

$$\left| B_0(m)\frac{1}{n}\sum_{i=1}^n \max_{j \in [m]}(N_{i,j}) \pm A_0(m) - B_0(m)\mathbb{E}_\pi \max_{j \in [m]}(N_{1,j}) \right| \leq B_0(m)B\sqrt{\frac{(1+\lambda)}{2n(1-\lambda)} \ln\left(\frac{2}{\delta}\right)} \tag{18}$$

$$\left| C_0(m, n) - A_0(m) - B_0(m)\mathbb{E}_\pi \max_{j \in [m]}(N_{1,j}) \right| \leq B_0(m)B\sqrt{\frac{(1+\lambda)}{2n(1-\lambda)} \ln\left(\frac{2}{\delta}\right)} \tag{19}$$

---

[5]We note that Fan et al. (2021) only present a one-sided bound, but by standard symmetry arguments this immediately implies the above two-sided bound.

Now we follow similar steps for $C_F(m,n)$. To start, we bound the distance from $\max_{j \in m} \frac{1}{n} \sum_{i=1}^{n} N_{i,j}$ and $\mathbb{E}_\pi[N_{11}]$ in terms of a sum of individual distances using the union bound.

$$\mathbb{P}_\pi \left( \left| \max_{j \in m} \frac{1}{n} \sum_{i=1}^{n} N_{i,j} - \mathbb{E}_\pi N_{11} \right| \geq \epsilon \right) = \mathbb{P}_\pi \left( \max_{j \in m} \frac{1}{n} \sum_{i=1}^{n} N_{i,j} - \mathbb{E}_\pi N_{11} \geq \epsilon \right)$$

$$+ \mathbb{P}_\pi \left( \max_{j \in m} \frac{1}{n} \sum_{i=1}^{n} N_{i,j} - \mathbb{E}_\pi N_{11} \leq -\epsilon \right) \tag{20}$$

$$= \mathbb{P}_\pi \left( \bigcup_{j=1}^{m} \left\{ \frac{1}{n} \sum_{i=1}^{n} N_{i,j} - \mathbb{E}_\pi N_{11} \geq \epsilon \right\} \right)$$

$$+ \mathbb{P}_\pi \left( \bigcup_{j=1}^{m} \left\{ \frac{1}{n} \sum_{i=1}^{n} N_{i,j} - \mathbb{E}_\pi N_{11} \leq -\epsilon \right\} \right) \tag{21}$$

$$\leq \sum_{j=1}^{m} \left[ \mathbb{P}_\pi \left( \frac{1}{n} \sum_{i=1}^{n} N_{i,j} - \mathbb{E}_\pi N_{11} \geq \epsilon \right) \right.$$

$$\left. + \mathbb{P}_\pi \left( \frac{1}{n} \sum_{i=1}^{n} N_{i,j} - \mathbb{E}_\pi N_{11} \leq -\epsilon \right) \right] \tag{22}$$

$$= \sum_{j=1}^{m} \mathbb{P}_\pi \left( \left| \frac{1}{n} \sum_{i=1}^{n} N_{i,j} - \mathbb{E}_\pi N_{11} \right| \geq \epsilon \right) \tag{23}$$

$$= m \mathbb{P}_\pi \left( \left| \frac{1}{n} \sum_{i=1}^{n} N_{i,1} - \mathbb{E}_\pi N_{11} \right| \geq \epsilon \right) \tag{24}$$

$$= m \mathbb{P}_\pi \left( \left| \sum_{i=1}^{n} N_{i,1} - n \mathbb{E}_\pi N_{11} \right| \geq n\epsilon \right) \tag{25}$$

Applying Proposition A.2 on the Markov Chain $(\boldsymbol{X}_{1,i}, N_{1,i})_{i \geq 1}$ with $f(\boldsymbol{X}_{1,i}, N_{1,i}) = N_{1,i} \in [0, B]$ and following the same steps as for $C_0(m,n)$, we similarly recover

$$\left| C_F(m,n) - A_F(m) - B_F(m) \mathbb{E}_\pi \max_{j \in [m]} (N_{1,j}) \right| \leq B_F(m) B \sqrt{\frac{(1+\lambda)}{2n(1-\lambda)} \ln \left( \frac{2m}{\delta} \right)} \tag{26}$$

which is the result in the Theorem. □

**Proposition A.3.** *Fix $m, K \in \mathbb{N} \backslash \{0\}$ and let $\mathbb{P}_N$ be a probability measure on $\mathbb{R}_+$ strictly positive first moment. Suppose (i) $N_1, ..., N_m \overset{iid}{\sim} \mathbb{P}_N$, (ii) $c_1(m), ..., c_K(m) \geq 0$ and (iii) $\alpha \in [\max_{j \in [K]} c_j(m) / \sum_{j \in [K]} c_j(m), 1]$. Then, we have*

$$E(m) := \frac{c_{\neg k}(m) + c_k(m) \mathbb{E} \max_{j \in [K]} N_j}{\alpha(c_{\neg k}(m) + c_k(m))(K - 1 + \mathbb{E} N_1)} \leq \frac{\mathbb{E} \max_{j \in [K]} N_j}{\mathbb{E} N_1} =: R(m) \tag{27}$$

*where $c_{\neg k}(m) = \sum_{j \neq k} c_j(m)$. The bound is tight.*

*Proof.* Note $\frac{a+b}{c+d} = \frac{a}{c}\gamma + \frac{b}{d}(1 - \gamma)$ where $\gamma = \frac{c}{c+d}$ for any $a, b, c, d \in \mathbb{R}$. Applying this to our case, we get

$$E(m) = \frac{c_{\neg k}(m)}{\alpha(c_{\neg k}(m) + c_k(m))(K-1)} w + \frac{c_k(m)}{\alpha(c_{\neg k}(m) + c_k(m))} R(m)(1 - w) \tag{28}$$

where $w = \frac{\alpha(c_{\neg k}(m) + c_k(m))}{\alpha(c_{\neg k}(m) + c_k(m))(K - 1 + \mathbb{E} N_1)} \in [0, 1]$. Now we split into two cases for $K = 1$ and $K > 1$. For the case $K = 1$ we only have a single iterative state and so $C_{\neg k}(m) = 0$, $\alpha = 1$. In this case we trivially have $E(m) = R(m)$. Now

suppose $K > 1$. In this case, since $\alpha(c_{\neg k}(m) + c_k(m)) \geq \max_{j \in [K]} c_j(m)$, we have

$$\frac{c_k(m)}{\alpha(c_{\neg k}(m) + c_k(m))} \leq \frac{c_k(m)}{\max_{j \in [K]} c_j(m)} \leq 1 \tag{29}$$

Which means we can bound $E(m)$ by removing the term in front of $R(m)$,

$$E(m) \leq \frac{c_{\neg k}(m)}{\alpha(c_{\neg k}(m) + c_k(m))(K-1)} w + R(m)(1-w) \tag{30}$$

By the same logic, we have

$$\frac{c_{\neg k}(m)}{\alpha(c_{\neg k}(m) + c_k(m))(K-1)} \leq \frac{c_{\neg k}(m)}{\max_{j \in [K]} c_j(m)(K-1)} = \frac{\sum_{j \neq k} c_j(m)}{\max_{j \in [K]} c_j(m)(K-1)} \leq 1 \tag{31}$$

which means

$$E(m) \leq w + R(m)(1-w) \tag{32}$$
$$\leq R(m) \tag{33}$$

Where the last line uses the fact that $R(m) \geq 1$ since $N_1 \geq 0$. The bound is tight because when $K = 1$ we have $E(m) = R(m)$. This completes the proof

$\square$

# B. Additional Details

## B.1. Algorithms

Note here we use $\tilde{x}$ to denote a batch of inputs $[\boldsymbol{x}_1, ..., \boldsymbol{x}_m]$ for $m$ different chains, and the same for other variables.

---

**Algorithm 6** Vectorized MCMC algorithm with `vmap(sample)` function

---

1: **Inputs**: sample $\tilde{x}_0$, seed $\tilde{r}_0$
2: **for** $i \in \{1, ..., n\}$ **do**
3:  generate $\tilde{x}_i, \tilde{r}_i \leftarrow$ `vmap(sample)` $(\tilde{x}_{i-1}, \tilde{r}_{i-1})$
4: **end for**
5: **return** $\tilde{x}_1, \ldots, \tilde{x}_n$

---

**Algorithm 7** Vectorized FSM MCMC algorithm with `vmap(step)` function

---

1: **input:** initial value $\tilde{x}_0$, # samples $n$
2: **initialize:** $\tilde{z} =$ `vmap(init)` $(\tilde{x}_0), \tilde{X} = \text{list}(), \tilde{B} = \text{list}()$
3: Set $\tilde{t} = 0$ and $\tilde{k} = 0$
4: **while** $\min_{\tilde{t}_i \in \tilde{t}} \{\tilde{t}_i\} < n$ **do**
5:  $(\tilde{k}, \tilde{z}, \text{isSample}) \leftarrow$ `vmap(step)` $(\tilde{k}, \tilde{z})$
6:  append current sample value $\tilde{x}$ stored in $\tilde{z}$ to $\tilde{X}$
7:  append `isSample` to $\tilde{B}$
8:  update sample counter $\tilde{t} \leftarrow \tilde{t} + \text{isSample}$
9: **end while**
10: **return** $\tilde{X}[\tilde{B}]$

---

**Algorithm 8** Transition kernel for elliptical slice sampler with log-pdf $\log p$, covariance matrix $\Sigma$.

---

1: **Input:** Sample $\boldsymbol{x}$
2: Choose ellipse $\boldsymbol{\nu} \sim \mathcal{N}(\mathbf{0}, \Sigma)$
3: Set threshold $\log y \leftarrow \log p(\boldsymbol{x}) + \log u : u \sim \mathcal{U}[0, 1]$
4: Set bracket $[\theta_{\min}, \theta_{\max}] \leftarrow [\theta - 2\pi, \theta] : \theta \sim \mathcal{U}[0, 2\pi]$
5: Make proposal $\boldsymbol{x}' \leftarrow \boldsymbol{x} \cos \theta + \boldsymbol{\nu} \sin \theta$
6: **while** $\log p(\boldsymbol{x}') > \log y$ **do**
7:  Shrink bracket and update proposal:
8:  **if** $\theta < 0$ **then**
9:   $\theta_{\min} \leftarrow \theta$
10:  **else**
11:   $\theta_{\max} \leftarrow \theta$
12:  **end if**
13:  $\boldsymbol{x}' \leftarrow \boldsymbol{x} \cos \theta + \boldsymbol{\nu} \sin \theta : \theta \sim \mathcal{U}[\theta_{\min}, \theta_{\max}]$
14: **end while**
15: **Return** $\boldsymbol{x}'$

---

**Algorithm 9** `bundled_step` as input to `amortized_step`.

---

1: **Input:** Algorithm state $k$, variables $z$
2: `doComputation` = False
3: **while not** `doComputation` **do**
4:  $(k, z, \text{isSample}, \text{doComputation}) \leftarrow$ `step(k, z)`
5: **end while**
6: **Return** $(k, z, \text{isSample}, \text{doComputation})$

---

## B.2. FSM Construction Procedure for Programs Considered in Main Text

**Detailed Construction in the two sequential while loops case**  Here, we describe the FSM construction for programs with two sequential while loops in detail. Following the constructions introduced in the main body, let us note

- $\mathcal{F}_1 \coloneqq (\{S_{11}, S_{12}, S_{13}\}, \mathcal{Z}, \delta_1, S_{11}, S_{13})$ the FSM associated to $B_1$

- $\mathcal{F}_2 \coloneqq (\{S_{21}, S_{22}, S_{23}\}, \mathcal{Z}, \delta_2, S_{21}, S_{23})$ the FSM associated to $B_2$

By construction, $S_{21}$ is empty. Both FSMs share the same input space, which is the set of local variables values associated to the original `sample` function.

Then, the resulting FSM representation of `sample` is $(\mathcal{S}, \mathcal{Z}, \delta, S_{11}, S_{23})$, where

- $\mathcal{S} = \{S_{11}, S_{12}, S_{13}, S_{22}, S_{23}\}$.

- Transition function $\delta$ defined as

$$\delta(S, z) = \begin{cases} \delta_1(S, z) & \text{if} \quad S \in \{S_{11}, S_{12}\} \\ \delta_2(S_{21}, z) & \text{if} \quad S = S_{13} \\ \delta_2(S, z) & \text{if} \quad S = \{S_{22}, S_{23}\}, \end{cases} \tag{34}$$

whose construction illustrates the "FSM stitching" operation performed.

**Detailed Construction in the two nested while loop case**  Here, we describe the FSM construction for programs with two nested while loops. Following the constructions introduced in the main body, let us note

$\mathcal{F}_i \coloneqq (\{S_{i1}, S_{i2}, S_{i3}\}, \mathcal{Z}, \delta_i, S_{i1}, S_{i3})$ the (inner) FSM associated to $B_2$.

Then, the resulting FSM representation of `sample` is $(\mathcal{S}_o, \mathcal{Z}, \delta_o, S_1, S_3)$, where

- $\mathcal{S}_o = \{S_1, S_{i1}, S_{i2}, S_{i3}, S_3\}$.

- Transition function $\delta_o$ defined as

  - $\delta_o(S_1, z) = \delta_o(S_{i3}, z)$ runs the outer while loop condition on $z$, goes to $S_{i1}$ if True, and $S_3$ otherwise.
  - $\delta_o(S, z) = \delta_i(S, z)$ if $S \in \{S_{i1}, S_{i2}\}$

## B.3. FSM Construction Procedure for General Programs.

The FSM construction procedures in **??** for specific programs can be automated to handle programs with any finite number of sequential and/or nested while loops. We show how this works using a simple programming language defined by the following grammar:

$$
\begin{aligned}
\text{Operation op} &::= & \text{op}_1 \mid \text{op}_2 \mid \cdots \mid \text{op}_n \\
\text{Simple Block SB} &::= & \langle \text{op} \rangle \mid \langle \text{op} \rangle \, \langle \text{SB} \rangle \\
\text{While Block WB} &::= & \text{W} \, \{ \, \langle \text{Bs} \rangle \, \} \\
\text{Block B} &::= & \langle \text{SB} \rangle \mid \langle \text{WB} \rangle \\
\text{Blocks Bs} &::= & \langle \text{B} \rangle \mid \langle \text{Bs} \rangle \, \langle \text{B} \rangle
\end{aligned}
$$

This grammar contains just enough to describe non-trivial programs containing any finite number of (possibly nested) while loops. Given a program generated with this grammar, we construct an FSM variant of it by:

1. Parsing the program into a parse tree.

2. "Coarsening" the parse tree into a tree where each node is a block.

3. Applying a recursively-defined function which transforms the program into an FSM.

4. Collapsing any spurious empty states.

B.3.1. STEP 1: PARSING THE PROGRAM INTO A PARSE TREE.

First, note that as this grammar is context-free, every program $P$ can be represented by the yield of a parse tree (e.g. Theorem 5.12 (Hopcroft et al., 2001)). Consequently, one can use any context-free parser to obtain a parse tree of the program. We choose the look-ahead, left-to-right, rightmost derivation (LALR) parser (DeRemer, 1969). Figure 8 shows the resulting parse tree associated with the program "op1W{op1W{op1}}".

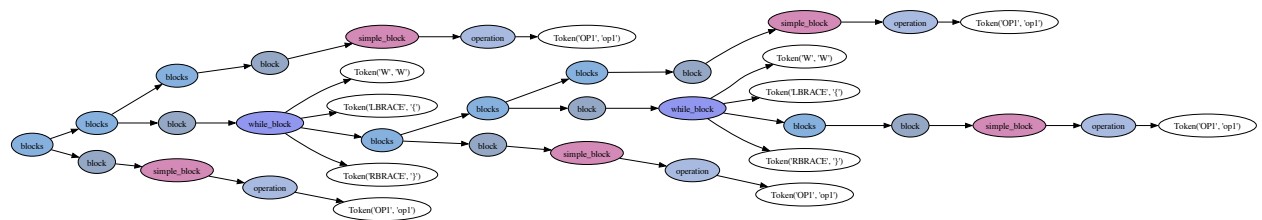

*Figure 8.* Parse tree of the program "op1W{op1W{op1}}".

B.3.2. STEP 2: COARSENING THE PARSE TREE.

To convert the parse tree into an FSM, we first derive a simplified version of it with: (i) no operation nodes (e.g. content of simple blocks), (ii) no block markers (e.g. While tokens and braces), (iii) no "blocks" and "block" nodes (these are helpful to clarify the grammar's definition, but do not carry information about a given program). The first two properties can be achieved by simply pruning the tree of all nodes that are not blocks, which is a well-known operation and not shown here. The output of this procedure is guaranteed to be a tree, as all such nodes (i) and (ii) are terminal. The second property can be achieved by applying the procedure in Algorithm 11.

---

**Algorithm 10** Gathering meaningful children

1: **function** GATHER_CHILDREN(*node*) **returns** list of Node
2:     all_children ← ∪ MAP(GATHER_CHILDREN, node.children)
3:     all_children ← ∪ FILTER(NON_TERMINAL, all_children) {Discard "op" nodes and markers}
4: **if** node.type is "block" or "blocks" **then**
5:     **return** all_children {Discard the node, and only return the children}
6: **else**
7:     **return** [Node(node.type, all_children)]
8: **end if**
9: **end function**

---

**Algorithm 11** Coarsening the parse tree

1: **function** COARSEN_TREE(*node*) **returns** Node
2:     all_children ← ∪ MAP(GATHER_CHILDREN, children(node))
3:     **return** Node("blocks", all_children)
4: **end function**

---

After applying the coarsening procedure to the parse tree of the same program "op1W{op1W{op1}}", we obtain the coarsened parse tree shown in Figure 9. The algorithm has been modified to return block labels in order to track the graph manipulation operations done in the next steps.

B.3.3. STEP 3: TRANSFORMING THE COARSENED PARSE TREE INTO AN FSM.

Now we transform the coarsened parse tree into an FSM. Our approach leverages the fact that any subtree of the parse tree is itself a valid (sub)program[6], and should thus be handle-able by the conversion procedure we define. Each (sub)program

---

[6]It is a "subprogram" of the original program in the sense that it is contained in the original program.

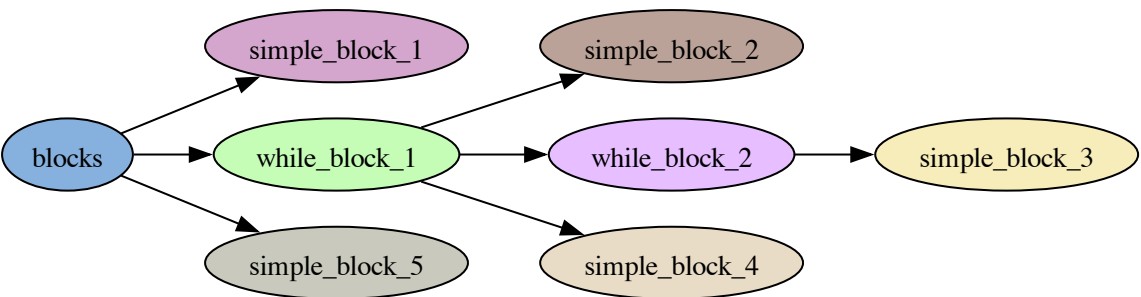

*Figure 9.* Coarsened parse tree of the program "op1W{op1W{op1}}".

is either a single simple block, an arbitrary while block (looping over a sequence of blocks), or the root program (e.g. a sequence of blocks).

This suggests a recursive conversion procedure which computes a single-state FSM for the block at the leaf nodes, and then (recursively) combines these FSMs into larger FSMs at each layer of the parse tree, until we reach the root node, at which point the final FSM is returned. The FSM at leaf nodes (i.e. simple blocks) is a simple single state FSM. The FSM at non-leaf nodes (e.g. while blocks or the entire program) is constructed by (i) connecting the terminal state of the $i^{th}$ child's FSM to the initial state of the $(i+1)^{th}$ child's FSM, for every child $i$ of the non-leaf node except the last, and (ii) if the node is a while block, (a) adding an edge from the last child's FSM to the first child's FSM, and (b) assigning the initial and terminal state of the current node's FSM to empty states with an edge between them. Additions (a) and (b) account for the fact that unlike standard blocks, while blocks have bodies that may be looped over, or skipped.

Using empty states as initial and terminal FSM of while blocks allows our FSM conversion procedure to be "context-free"—each subFSM is obtained by assembling sub-subFSMs, without knowledge about the nature of the blocks being connected. Without these empty states, we would require more complicated logic to perform the connection. For instance, in a program of the form "op1W{op1}W{op1}op2", one needs to connect the first op1 to the last op2 to encode the fact that both while loops may be skipped.

The full algorithm is given in Algorithm 12. The resulting FSM for the program "op1W{op1W{op1}}" is shown in Figure 10. The nodes starting with an "I" (resp. "T") are the (empty) initial (resp. terminal) states of each while block. The transitions are labeled using the following convention:

- "E$\langle n \rangle$" means "enter loop number $n$"

- "C$\langle n \rangle$" means "continue loop number $n$"

- "S$\langle n \rangle$" means "skip loop number $n$"

- "F$\langle n \rangle$" means "exit loop number $n$"

- "N" means "next block" (which are created when connecting the different subFSMs of the root program)

B.3.4. STEP 4: COLLAPSING THE EMPTY STATES.

The empty states produced by the step above are side effects of the conversion procedure used: the final FSM should not contain them. To this end, we next describe a post-processing, iterative procedure which removes them from the FSM. At each iteration, the procedure selects (if any) an empty state, and—unless it transitions into itself—(i)removes from the list of FSM nodes and (ii) adds edges connecting the parents of the empty state to its children.

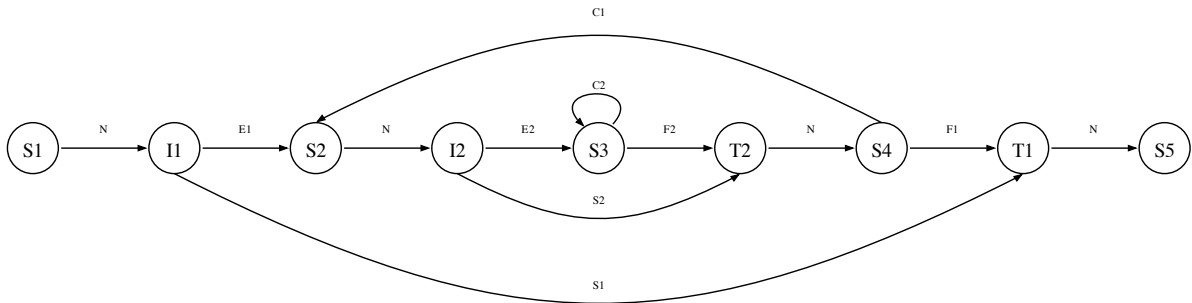

*Figure 10.* Intermediate FSM obtained from the coarsened parse tree of the program "op1W{op1W{op1}}". This intermediate FSM contains empty states, which are collapsed to obtain the final FSM.

---

**Algorithm 12** FSM creation from a coarsened parse tree

---

1: **function** COARSE_TREE_TO_FSM(*node*) **returns** FSM
2: **if** node.type is "while_block" or "blocks" **then**
3:     c_fsms ← MAP(COARSE_TREE_TO_FSM, node.children)
4:     nodes ← ∪ MAP(GET_NODES, c_fsms)
5:     edges ← ∪ MAP(GET_EDGES, c_fsms)
6:     labels ← ∪ MAP(GET_LABELS, c_fsms)
7:     // Connect the different subFSMs
8:     conn_edges ← {(c_fsms[i-1].terminal, c_fsms[i].initial) for $i = 1, \ldots, \text{len}(c\_fsms) - 1$}
9:     conn_labels ← {"N", . . . , "N"}
10:     edges, labels ← edges ∪ conn_edges, labels ∪ conn_labels
11:     i ← COUNTER()
12: **if** node.type == "while_block" **then**
13:         terminal_state, initial_state ← Node("terminal"), Node("initial")
14:         enter_loop_edge ← (initial_state, c_fsms[0].initial)
15:         exit_loop_edge ← (c_fsms[-1].terminal, terminal_state)
16:         continue_loop_edge ← (c_fsms[-1].terminal, c_fsms[0].initial)
17:         skip_loop_edge ← (terminal_state, initial_state)
18:         nodes ← nodes ∪ {terminal_state, initial_state}
19:         edges ← edges ∪ {continue_loop_edge, skip_loop_edge, enter_loop_edge, exit_loop_edge}
20:         labels ← labels ∪ {"E" + i, "F" + i, "C" + i, "S" + i}
21:         **return** FSM(nodes, edges, labels)
22:     **else**
23:         **return** FSM([simple_block], [], [])
24:     **end if**
25: **end if**
26: **end function**

---

The labels of the new edges are obtained by concatenating the "parent-to-empty state" and "empty state-to-children" edges. The procedure is given in Algorithm 13, and the resulting FSM for the program "op1W{op1W{op1}}" is shown in Figure 11. The procedure results in FSMs that agree those in the main text for single while loops, two nested while loops, and two sequential while loops. There are two minor caveats, which we discuss below.

*Remark* B.1 (Handling empty states with self-transitions). The procedure above removes all empty states, apart from those with self-transitions. These self-transitions are not present before collapsing the empty states, and arise in the midst of the procedure, after collapsing a subset out of all the FSM empty state. Importantly, such self-transitions are pathological: as the state of the program does not change when performing a self-transition into an empty node, if such a self-transition occurs, it must keep occurring indefinitely afterwards. Thus, programs resulting into FSM with self-recurring empty states form a class of "potentially non-halting" programs.

*Remark* B.2 (Handling impossible transitions). The node collapsing process results in the creation of "multistep" edges. These transitions involve evaluating multiple conditions in a single step. Sometimes, these conditions are mutually exclusive: in that case, this multistep transition is impossible, and can be removed from the graph. By removing such transitions after each step of the procedure, the routine can be shown to not add any duplicated edges to the FSM, as proved in the next proposition.

---

**Algorithm 13** Collapsing empty states

---

 1: **function** COLLAPSE_EMPTY_STATES(fsm) **returns** FSM
 2:     edges ← fsm.edges
 3:     labels ← fsm.labels
 4: **for all** node **in** FILTER(EMPTY | NOT INITIAL | NOT FINAL, fsm.nodes) **do**
 5:     **if** node ∈ node.children **then**
 6:         **continue**
 7:     **end if**
 8:     **for all** parent, child **in** PRODUCT(node.parents, node.children) **do**
 9:         new_edge ← (parent, child)
10:         **if** new_edge ∈ edges **then**
11:             **error** {Edge already exists}
12:         **end if**
13:         new_label ← AND(parent.label, child.label)
14:         **if** IMPOSSIBLE(new_label) **then**
15:             **continue**
16:         **end if**
17:         edges ← edges ∪ {new_edge}
18:         labels ← labels ∪ {new_label}
19:     **end for**
20: **end for**
21:     nodes ← FILTER(NON_EMPTY, fsm.nodes)
22:     **return** FSM(nodes, edges, labels)
23: **end function**

---

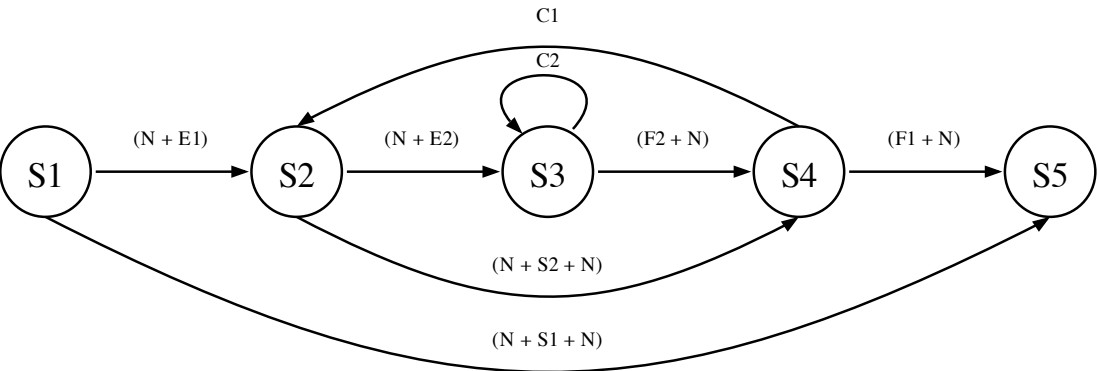

*Figure 11.* FSM for the program "op1W{op1W{op1}}", with empty states collapsed, using the automatic procedure. Note that this matches the structure of the FSM for the No-U-Turn sampler in Figure 2, which was produced using (essentially) a special case of this procedure for programs with two nested while loops.

## B.4. Implementation Details

**FSM Step Function Design.** At each step of the FSM, one always executes a block $S_k : z \mapsto z'$ which updates the inputs, before calling the transition function $\delta$ on $(k, S_k(z))$ to determine which block should be run next. One can therefore combine these into a single (state) function $T : (k, z) \mapsto (\delta(k, S_k(z)), S_k(z))$, or equivalently, a collection of functions $T_1, ..., T_K$ where $T_k : z \mapsto (\delta(k, S_k(z)), S_k(z))$. In practice, our implementation of Algorithm 2 calls a single `switch` over these composite state functions to streamline the code. The same composite state functions are also used in our implementation of `bundled_step` in Algorithm 4.

**FSM Runtime Design.** In our experiments, we use a native Python while loop in Algorithm 3 (and its `vmap`-ed variant in Algorithm 7). Inside the loop, we use a (JIT compiled) `jax.lax.scan` to run the FSM step function for blocks of $t = 100$ steps, when drawing $n > 100$ samples. This gives us the flexibility to store the results in dynamically shaped lists/arrays and transport to the CPU for faster array slicing when CPU memory is available, whilst still reaping the benefits of JIT compilation.

**Compilation.** We JIT compile both `vmap(step)` and `vmap(sample)` functions for each MCMC algorithm implementation (here `step` refers to any of the basic variant Algorithm 2, bundled variant Algorithm 4, amortized variant Algorithm 5, or the bundled and amortized variant Algorithm 9). For Delayed Rejection and the Elliptical Slice Sampler (with $n = 25$) we remove compilation time to get more accurate results or the runs with small numbers of chains $m$, due to the low cost of the computations involved.

**JAX implementation.** When comparing to non-FSM implementations, we used BlackJAX (Cabezas et al., 2024) for fair comparison with our method, since we use BlackJAX primitives for the key computations in some of our algorithms (e.g. HMC-NUTS). Where a BlackJAX implementation was not available (e.g. Delayed Rejection), we wrote our own for fair comparison with our FSM implementation.

**Bundling with Amortized Step.** As discussed in Section 5, one cannot non-trivially use `bundled_step` as the step function inside `amortized_step`, because a given sequence of states may require the amortized computation to be updated in the middle, and nothing in `bundled_step` flags this. To reap some of the benefits of step bundling when amortizing an expensive computation $g$ (e.g. expensive log-pdf calculations), we use another step function defined in Algorithm 9, as the 'step' function called inside `amortized_step`. This step function iteratively runs `step` (modified to return the `doComputation` flag) until `doComputtion = True` - ironically, using a while loop. When `vmap`-ed, each 'step' inside `amortized_step` now executes sequences of cheap states that do not require $g$, until $g$ is required again for all chains. This works effectively when $g$ is called inside the iterative state, which is typically the case when $g = \log p$.

## B.5. Additional Results

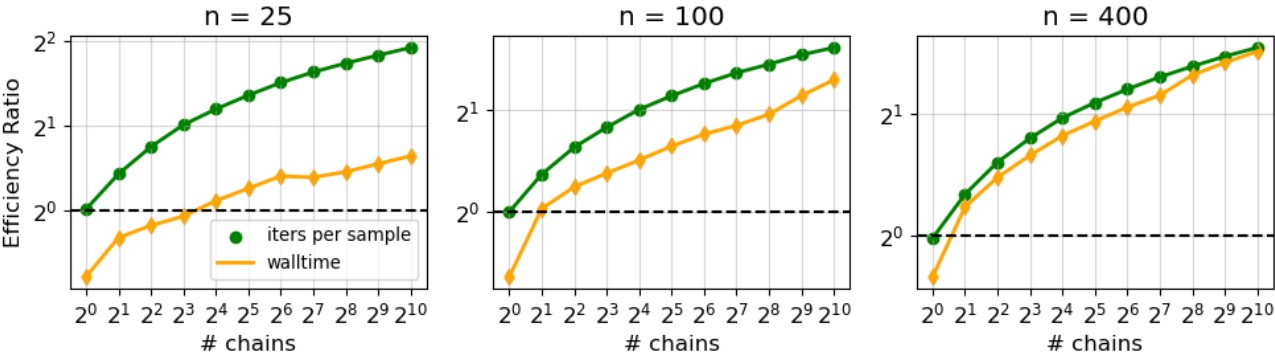

*Figure 12.* Efficiency Ratio of our elliptical slice FSM against BlackJAX's elliptical slice algorithm (as measured by estimated $R(m) = \mathbb{E}[\max_{j \in [m]} N_j] / \mathbb{E}[N_1]$ (i.e. iters per sample) and walltime) on the Real Estate Dataset described in Section 7 when restricting the dataset to the first $n \in \{25, 100, 400\}$ datapoints. The relative efficiency of the FSM improves as the number of chains used increase, and as the log-likelihood cost increases. When $n = 400$, we almost achieve the theoretical bound $R(m)$ in speed-ups.

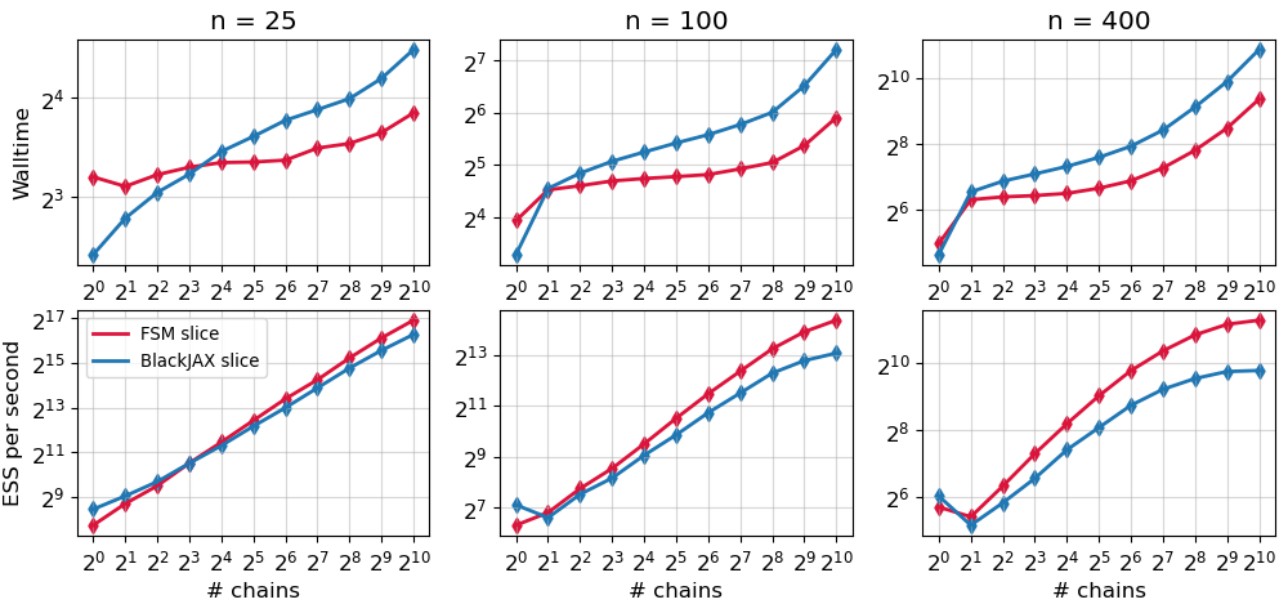

*Figure 13.* Walltimes and ESS per second using the Elliptical Slice Sampler (non-FSM vs FSM implementation) on the Real Estate Dataset described in Section 7, when restricting the dataset to the first $n \in \{25, 100, 400\}$ datapoints. For each dataset size, the best walltime and ESS/second is obtained by both implementations when using $m = 1024$ chains. Our FSM implementation can obtain the greatest efficiency for all dataset sizes. As the log-likelihood cost increases (the log-likelihood in GPR regression costs $\mathcal{O}(n^3)$), we see the FSM efficiency gain increase, reflecting the benefits of amortization.

| Distribution | NUTS-FSM | Standard NUTS | FSM Speed-up |
|---|---|---|---|
| Real-Estate GPR | 863.6 | 274.1 | 3.15x |
| Soil | 0.014 | 0.006 | 2.43x |
| Pilots | 3.538 | 3.869 | 0.91x |

*Table 2.* ESS/sec comparison for NUTS with and without FSM acceleration on Real-Estate GPR sampling problem considered in Section 7.2, and two sampling problems (Soil and Pilots) from the PosteriorDB database (Magnusson et al., 2025). Results are averaged over 1000 samples and 128 chains. As with Section 7.4 in the main text, we use 400 warm-up steps to tune the mass matrix and step size of our FSM implementation, and the BlackJAX (non-FSM) implementation. We find substantial speed-ups in 2/3 problems. Both of these problems (Real-estate GPR and Soil) have an expensive log-pdf (and gradient) whilst the log-pdf in Pilots is much cheaper. In general, when the number of integration steps taken by NUTS is moderate to high on average (so that most time is spent integrating along the trajectory), an expensive log-pdf makes the additional cost of executing all other blocks in the FSM step function (when integrating) relatively smaller. That is, the gap between $E(m)$ and $R(m)$ (as defined in Section 4) is smaller, hence increasing the relative efficiency of the FSM. The opposite happens when the log-pdf is cheap (as in Pilots). We also note that computational resources did not balance out across chains in Soil and Pilots after 1000 samples. We therefore would expect to see larger speed-ups when running for longer.

