# OpenReview forum: "Efficiently Vectorized MCMC on Modern Accelerators"
_ICML.cc/2025/Conference — ICML 2025 spotlightposter_

### Official Review · Reviewer_3kkK · 2025-03-02

**Overall Recommendation:** 4

**Summary:**

With the advancement of AI infrastructures, it is increasingly interesting to scale algorithms up with parallelism. Nevertheless, it is not efficient to run parallel MCMC with naive automatic vectorization (e.g., vmap), due to a varying execution time at each sampling step across different chains. The current approach will wait for the slowest chain at each step, while this paper proposes to remove this synchronization problem (usually caused by while-loops) by rewriting an MCMC algorithm as transitions in a finite state machine (FSM). With this modification, we can run parallel MCMC as parallel transition functions in the FSM, where different chains could be in different states, and synchronization will only happen once in the end. On the other hand, the proposed FSM based parallel MCMC also introduces additional control flow cost because codes for all states have to be executed every step no matter what state a chain is in.

Based on the idea, this paper develops a theoretical framework for analysing such parallel MCMC algorithms. Namely, the expected running time of FSM v.s. standard parallel MCMCs will depend on the number of chains, the number of samples per chain, the convergence of the execution time per step, the "organization" of the FSM, etc. The takeaway is that the running time of standard parallel MCMC will converge to the slowest chain while the running time of FSM parallel MCMC will converge to a single chain scaled by the additional cost of control flow.

To reduce the cost of control flow, the paper proposes two techniques, called step bundling and cost amortization. Step bundling allows transitions to be combined in some cases. Cost amortization makes sure that an expensive function (usually the log density function) will only be executed once per FSM step.

The experiments test different aspects of the FSM parallel MCMC. FSM is the most helpful when the distribution of one-step sampling time is skewed. When implemented, FSM easily outperforms standard parallel MCMC with broad implementations of Delayed Rejection MH, elliptical slice sampling, HMC-NUTS and transport elliptical slice sampling, when the # of chains is sufficient. In addition, the control flow optimization is helpful on top of FSM and may introduce substantial speedups.

**Claims And Evidence:**

The empirical evidences are strong but the theoretical evidences have several presentation issues. However, I do believe all claims are correct. See below for details.

**Essential References Not Discussed:**

No.

**Experimental Designs Or Analyses:**

I checked the settings of all experiments and they all look reasonable to me.

-  Among all the MCMC algorithms, I am using NUTS the most. The NUTS experiment in 7.3 assumes identity mass matrix which looks different from standard approaches. But it is chosen to demonstrate how FSM helps the synchronization problem. However, I believe there should be more experiments on NUTS for a paper like this. It would be much more interesting to try NUTS on the models of Table 1.

**Methods And Evaluation Criteria:**

The proposed method solves a key problem in efficient MCMC sampling. The problem of synchronization is part of why MCMC were mainly run with few chains on small machines. The solution of this work is neat and makes sense from the perspective of parallel computing systems. I would love to see the solution to be implemented in every modern MCMC libraries.

There are two axes of the benchmark: the MCMC algorithms and the models. The work has a broad selection of benchmark MCMC algorithms (delayed rejection MH, elliptical slice sampling, HMC-NUTS and transport elliptical slice sampling), which are all impactful in practice. The benchmark models include both simulated and practical models, covering difficult cases such as high correlations and funnels. I especially like the arguments of acceptance rate of at least one chain in parallel MCMC (line 436), which highlights the necessity of FSM based approaches.

**Other Comments Or Suggestions:**

A list of typos I can find:

- Line 121 right: $k\in\\{0,1\\}$ -> $k\in\\{1,2\\}$
- Line 159 right: $k$ starts from 1 so probably $(1,z_1)$
- Line 212 right: $(n,m)$ and $(m,n)$ are used inconsistently
- Line 8 Algorithm 5: should $g\\_flag$ be returned?

**Other Strengths And Weaknesses:**

I believe there should be another change of mind about parallel MCMC following this work. The synchronization problem caused some chains to run "in vain" waiting for the slowest chain. With the FSM design, however, the waiting by the faster chain IS useful for sampling. For example, if we set $n=1,000$, the slowest chain will generate $1,000$ samples while the fastest chain might have generated $1,100$ samples. They shall all be mixed together for estimation. In practical settings, we can set a limit of # of transition cycles instead of # of samples to make more sense.

**Questions For Authors:**

My points can be condensed into the questions:

- Is there a more intuitive justification of (4), as well as (6) and (7)?
- How good is the performance of NUTS-FSM compared to NUTS on the models in Table 1?

**Relation To Broader Scientific Literature:**

As the paper also points out, the synchronization problem is well documented in the literature (BlackJax,2019;Sountsov et al.,2024;Radul et al.,2020). The paper borrows classic ideas of finite state machines from computer systems (Hopcroft et al.,2001), and shapes it with problems of MCMC sampling.

**Theoretical Claims:**

My major issue with this paper is about the theoretical framework. I believe they are correct, but there are also some obstacles that keep it from being checked thoroughly.

- Equations (3) and (4) work with expectations. I do not see how (4) is true given the information up until this part. More speficifically, my expression for (4) is $\mathbb{E}[\max_{j\le m}\sum_{i=1}^nN_{\infty,j}^{(i)}]$. The gap might be justified with Theorem 4.1, but some connections need to be made.

- The work refers to Algorithm 1 as "standard design" and Algorithm 3 as "FSM design". Both algorithms are analyzed based on the executing time of each state function in the FSM design. There is a disconnection between the state functions and Algorithm 1 as there are no states in Algorithm 1. To make sense, I am expecting an explanation of how much a step in Algorithm 1 costs as a function of $c_1, c_2,..., c_K$.

- Following the above point, I find the notations around line 220 confusing. $c_{\neg k}$ is not defined, and even if it is, I am not expecting $A_0$ to depend on a specific $k$. The presentations of $A_0, B_0, A_F, B_F$ are all not justified.

---

> ### Author Rebuttal · Authors · 2025-03-31
>
> Thank you for your diligent review - we are glad you believe our method is a worthwhile addition to modern MCMC libraries. Below we address your questions.
>
> **Theoretical framework:** Eq(3) and eq(4) hold without taking expectations under appropriate theoretical conditions, but we appreciate the symbol is doing a lot of heavy lifting in our effort to convey the basic ideas. With more standard probability theoretic notation, the exact form of eq(3) is $$\sum_{i=1}^n\max_{j \leq m}N_{i,j}=n\mathbb E\max_{j \leq m}N_{\infty,j} + \mathcal O_p(\sqrt{n})$$ which holds under an appropriate Central Limit Theorem or concentration inequality on the sequence $(\max_{j \leq m}N_{i,j} : i \geq 1)$. Since $0 < n \mathbb E\max_{j \leq m}N_{\infty ,j}  = \mathcal O(n)$ in our setting, as $n$ grows the stochastic error term becomes negligible.
>
> For eq(4), the exact version under equivalent conditions is
> $$\max_{j \leq m}\sum_{i=1}^nN_{i,j}=  \max_{j \leq m} n \mathbb EN_{\infty,j} + \mathcal O_p(\sqrt{n})$$
> which holds using the same arguments as eq(3) for the sum inside the max above, and using the fact that the max over finitely many $\mathcal O_p(\sqrt{n})$ variables is $\mathcal O_p(\sqrt{n})$.
>
> This is all made rigorous in Theorem 4.1, where we use an approach based on Hoeffding bounds for Markov chains.
>
> We appreciate the reviewer bringing this to our attention. We are happy to replace eq(3) and eq(4) by their reformulations above and modify the text accordingly.
>
> **Section 4 explanation:** We agree that more details on how the quantities in Section 4 hold would improve clarity.
>
> The states $S_1,...,S_K$ exist in Algorithm 1 inside `sample` (recall that our FSM decomposes `sample` into contiguous states/blocks $S_1,...,S_K$ and defines `step` as a single call of one of these states). A single call of `vmap(sample)` for $m$ chains executes all $S_l$ (for $l \neq k$) once, and $S_k$ (the iterative state) $\max_{j \leq m}N_{i,j}$ times, where $N_{i,j} =$ \# while loop iterations to get  sample $i$ for chain $j$ and the max reflects that the while loop executes until it terminates for all chains.
> Assume the cost of state $S_j$, when executed for a batch of $m$ chains using `vmap`, is $c_j(m)$. In this case, the cost of calling `vmap(sample)` to get sample $i$ for all chains is $$\sum_{l \neq k}c_l(m) + c_k(m)\max_{j \leq m}N_{i,j}$$ Averaging this cost over $n$ samples gives eq(6) with $A_0(m) = c_{\neg k}(m)=\sum_{l \neq k}c_l(m)$ (we appreciate $c_{\neg k}$ should have been made clear) and $B_0(m) = c_k(m)$.
>
> For eq(7), we follow similar steps with the main difference that now `vmap(step)` executes every block for all chains. This means the cost of a single FSM step is $\sum_{j =1}^Kc_j(m)$. The overall cost of sample $i$ for chain $j$ is therefore $A_0(m) + B_0(m) N_{i,j}$, where $A_0(m) = (K-1)(\sum_{j =1}^Kc_j(m))$ and $B_0(m) = \sum_{j =1}^Kc_j(m)$. In the paper we multiply by $\alpha$ to later motivate amortization. Averaging over $n$ samples gives eq(7).
>
> **Additional NUTS Experiments** Since another reviewer asked us to implement on datasets in the `posteriordb` database, in the limited time we chose to implement NUTS on (i) the Real-Estate dataset (Experiment 7.2), (ii) two datasets from Experiment 7.4, and (iii) two known challenging `posteriordb` datasets (`pilots` and `soil`). We used 128 chains + 1000 samples per chain + 400 steps pre-tuning for step size + mass matrix via the window adaptation algorithm. To save space, below we report the improvement in ESS/Sec using the FSM:
>
> - Real Estate: 3.14x
> - Google Stock: 3.36x
> - Predator Prey: 1.55x
> - Soil: 2.42x
> - Pilots: 0.91x
>
> We find substantial speed ups in ⅘ cases, and larger speed-ups than the FSM variant of Ellip-slice/TESS across all datasets we implemented for both methods. We note that because these datasets have highly variable local geometries (except real-estate GPR), computational resources did not average out across chains after 1000 samples. We expect that when sampling even longer chains, these speed-ups would be even larger. The Pilots dataset has a relatively cheap log-likelihood and less variation in the number of integration steps used, and so the gains in avoiding synchronization costs are offset by the increased cost per block of the FSM.
>
> **Using extra samples**: We also had the idea to let chains that finish early collect more samples while waiting for other chains. The problem is that this biases the samples, as chains initialized in regions that are easier to sample from are over-represented. Importance weighting could resolve this for estimating expectations, but we leave this to future work. Enforcing n samples per chain still allows us to avoid the inefficiency from chains waiting at every iteration with `vmap`.
>
> **Typos**: Thank you for pointing those out, we will amend them in revisions. In line 8 of Alg 5 `g_flag` is not meant to be returned - the flag is an output of the state functions, and is only used in `amortized_step`.

---

> > ### Comment · Reviewer_3kkK · 2025-04-02
> >
> > I appreciate the additional clarifications (in retrospect I missed the definition of $k$ during reviewing) and NUTS experiments (the inclusion of adaptation schemes is really helpful). I raise my score to 4.

---

### Official Review · Reviewer_UagZ · 2025-03-11

**Overall Recommendation:** 5

**Summary:**

This work focuses on minimizing synchronization steps when vectorizing MCMC algorithms. Modern MCMC methods, such as NUTS, involve stochastic number of operations per Markovian transition, depending on the initial sample and random seed. This introduces unavoidable overhead when running multiple chains in a vectorized manner, as each transition step requires synchronization, forcing all parallel chains to wait for the slowest one to complete, leading to wasted computational resources.

Addressing this long-standing challenge is crucial for enabling MCMC algorithms to fully leverage modern hardware, such as GPUs. Most existing approaches tackle this problem by designing new MCMC algorithms that minimize random execution steps (e.g., CHEES). In contrast, this work provides a systematic implementation strategy for existing algorithms—such as NUTS and the slice sampler—using finite state machines on a single chain, ensuring that synchronization occurs only at the final transition step during vectorization.

The authors present theoretical results quantifying the expected performance gains compared to naïve vectorization and empirically demonstrate the effectiveness of their approach across multiple examples.

**Claims And Evidence:**

This work makes careful and well-supported claims regarding the expected improvements of the proposed implementation strategy. The authors substantiate their arguments with rigorous yet concise theoretical analysis and thorough empirical validation.

I find their reasoning compelling and fully agree with their conclusions.

**Essential References Not Discussed:**

Not to my awareness.

**Experimental Designs Or Analyses:**

The selection of metrics, choice of competitors, and Bayesian examples all seem well thought out and appropriate.

However, for a fundamental idea like this (which I hold in very high regard), I believe a more comprehensive empirical evaluation would further strengthen the work. I recommend conducting additional evaluations on a broader set of models from PosteriorDB or TensorFlow Probability’s Inference Gym to provide more extensive validation.

**Methods And Evaluation Criteria:**

yes, they make sense.

I personally think a better metric to compare the efficiency gain is the number of synchronization steps per sec or per 100 sample.  Though, this might not be straightforward to log in the run time.

Also, I'm bit surprised that one FSM relative efficiency gain is only 3/4 times faster than the naive vectorization (as shown in Fig 6); I'm expecting a much more significant difference. See the question section for details.

**Other Comments Or Suggestions:**

- This could be due to my personal taste or lack of expertise in this field. However, I would appreciate a pedagogical example (e.g., ellipitical slice sampler with two chains) showcasing step by step the difference in the synchronization betweem the naive vectorization and the FSM variant.   To buy more space, I personally think it's okay to move section 5 to apdx.

- Readers with less familarity with jax could benefit from a proper definition of the `switch` function in Algorithm 2.

**Other Strengths And Weaknesses:**

n/a

**Questions For Authors:**

- I'm bit surprised that one FSM relative efficiency gain is only 3/4 times faster than the naive vectorization (as shown in Fig 6) given that the drastic difference in iterations per sample.   Why doesn't walltime and ESS/sec doesn't scale in the same rate? (My guess is that for the walltime, it's dictated by the slowest chain, while iter/sample is controlled by the average speed across the chains.)

- In the begining of section 4, the authors mentioned about the limitation of FSM vectorization that all branches would have to evaluated.  If my understanding is correct, this would be the same for the naive vectorization as well? As long as one call `vmap` or `@jit`, all branches of control flows will be evaluated? So in my perspective, regardless of the number of steps needed to obtain n samples, the performance of FSM vectorization will be lower bounded by the naive vectorization?

**Relation To Broader Scientific Literature:**

This work provides a thorough literature review of existing efforts to enable MCMC on modern hardware. To my knowledge, it takes a completely different approach—one that should absolutely be incorporated into modern probabilistic programming languages (PPLs). Instead of modifying the algorithm itself, this work focuses on optimizing its implementation, making it the first to do so in this manner.

**Theoretical Claims:**

I checked the correctness of the proof. The theoretical claims operate under very reasonable assumptions, and the proof is sort eluded in the maintext.

---

> ### Author Rebuttal · Authors · 2025-03-31
>
> Thank you for your detailed and engaging review, we are excited that you see the value in our contribution. Below we address your comments and questions.
>
> **FSM Efficiency gain:** The main reasons that the improvement in walltimes don’t match iters/sample in Fig 6 are (i) as you guessed, the slowest chain is still substantially slower than the mean after n=1000 samples, and (ii) the log-likelihood is very cheap to compute in this experiment, so control-flow costs (which are large in NUTS) dominate. Since each FSM `step` executes all states (see clarification below on branch evaluation cost vs. naive vectorization), this makes the trajectory integration more expensive per step, even with step bundling. It should also be noted that $R(m)$ is around $20$ for $m=100$ chains, and we obtain a speed-up of around 10x, so the gap is actually not that large. For $m=500$ chains it is larger, but we suspect this is because of limited GPU capacity.
>
> **FSM evaluating all branches:** All branches of control-flows are indeed evaluated when vectorizing both the FSM and the naive implementation with vmap. However, this has different effects in each case due to the differing control-flow used “between” the blocks/states. The FSM `step` uses a switch over all code blocks $S_1,…,S_K$ that comprise `sample`. This means that a single call of `vmap(step)` has to execute all blocks for all chains, whichever block the chains are currently on. By contrast, in `sample`, the control-flow “between blocks” takes the form of (possibly multiple) while loops. In the case of a single while loop with blocks $S_1,S_2,S_3$, this results in the loop body ($S_2$) being evaluated for all chains until they have finished iterating. However when calling $S_1$ and $S_3$ with sample, only those blocks are executed.
>
> Essentially, we exchange the variance in execution time in `sample` (across chains), for a bias. As a result, when there is little/no variance in execution time (e.g. the while loop always terminates after 1 iteration), the basic FSM (without our optimizations) can perform worse than naive vectorization. The step-bundling and amortization procedures outlined in Section 5 are specifically designed to minimize this bias. For example, by using `bundled_step` (with blocks called in chronological order) the FSM would run in the same time as the naive vectorization if there is no waiting and the loop terminates in one iteration, because a single step now progresses all chains through $S_1 \to S_3$. Amortization enables functions which are executed in multiple blocks/states to only be called once per FSM step, therefore reducing the additional overhead. For the elliptical slice sampler, doing this for the logpdf can reduce the cost of executing all blocks almost back to the cost of a single block, whenever the logpdf is expensive enough to dominate computational cost.
>
> **posteriordb/InferenceGym:** This is a great suggestion. Another reviewer also asked for more experiments with NUTS. Given the limited time, we have now implemented NUTS on several other datasets from our paper, and two datasets from posteriordb (Pilots and Soil-Incubation), which were in the list of challenging problems for NUTS in Table 1 of [1]. For the posteriordb problems we re-defined the logpdfs in JAX, and used the data from the JSON files at the posteriordb repository. Below we present the improvements in ESS/sec when implementing NUTS with our FSM (vs. BlackJax NUTS) on these datasets:
>
> - (Experiment 7.2) Real Estate: 3.14x
> - (Experiment 7.4) Google Stock: 3.36x
> - (Experiment 7.4) Predator Prey: 1.55x
> - (posteriordb) Soil: 2.42x
> - (posteriordb) Pilots: 0.91x
>
> We find substantial speed ups in ⅘ cases, and larger speed-ups than the FSM variant of EllipSlice/TESS across all datasets we implemented for both methods. We note that because most of these datasets have highly variable local geometries, computational resources did not average out much across chains after 1000 samples. We expect that when sampling even longer chains, these speed-ups would be even larger. The Pilots dataset has a relatively cheap log-likelihood and less variation in the number of integration steps used, and so the gains in avoiding synchronization costs are offset by the increased cost per block of the FSM.
>
> **Pedagogical Example:** We agree that walking through the effect of synchronization and using our FSM on a simple example would add further clarity to the paper. We had wanted to do this in the submitted version but could not due to space constraints. We propose to use part of the additional page available to include this in Section 2 for the Elliptical Slice Sampler, and return to it again in Section 3 after presenting the FSM.
>
> **Switch Function Definition:** Thank you for the suggestion, we will include an explanation in the final version.
>
> [1] Magnusson et al. (2024).  “posteriordb: Testing, Benchmarking and Developing Bayesian Inference Algorithms”, arXiv preprint arXiv:2407.04967.

---

> > ### Comment · Reviewer_UagZ · 2025-04-05
> >
> > Thank you for the clarification. After reading the rebuttal and comments from the other reviewers, I decide to remain my positive opinion on this work.

---

### Official Review · Reviewer_pDxh · 2025-03-12

**Overall Recommendation:** 2

**Summary:**

Vectorizing Markov Chain Monte Carlo (MCMC) algorithms using vmap to create multi chain code results in a synchronization problem. That is all chains have to wait for the last chain to be completed. The paper proposes to solve this problem by using FSMs, finite state machines, to design single-chain MCMC algorithms in a way that can avoid synchronization barriers when vectorizing with vmap. The authors implemented popular MCMC algorithms like slice sampling, hmc-nuts, delayed rejection.It is possible to write single chain MCMC code and call vmap to turn it into vectorized, multi chain code that can run in parallel on the same processor. Vmap transforms every instruction in the sample into a corresponding instruction operating on a batch of inputs matrix-vector multiplication to vectorize the sample. These instructions are executed in lock-step across all chains. Those MCMC libraries adopted ML frameworks with automatic vectorization tools as their backend. Bottlenecks for these libraries happen at control flow, if there is a while loop all chains wait for the last chain. Paper is showing transforming MCMC algorithms into equivalent sampler that is avoiding mentioned synchronization problem; the novelty is at the approach of transforming MCMC algorithms into FSMs.These FSMs enable us to breakdown ‘sample’ into series of smaller steps that has minimal variance in execution time and avoids using while loops.

**Claims And Evidence:**

The claims in the paper are supported by experimental evaluations.

**Essential References Not Discussed:**

-

**Experimental Designs Or Analyses:**

The experimental analysis makes sense but it is not possible to validate the results.

**Methods And Evaluation Criteria:**

The proposed methods make sense for the targeted problem.

**Other Comments Or Suggestions:**

none

**Other Strengths And Weaknesses:**

Strengths:
- Sample to FSM approach is generalizable for various algorithms
-Time complexity of FSM and corresponding functions are given; comprehensive detailed mathematical formula and derivation is given in detail
-Various MCMC algorithms are used including Delayed-Rejection Metropolis Hasting, Elliptical Slice Sampling,HMC-NUTS their approach is tested on various different algorithms

Weaknesses and Suggestions:
-The paper shows FSM of an MCMC algorithm with a single while loop, two sequential while loops, two nested while-loops. It does not mention different configurations and how FSM should be obtained if these while loops occur in different way or interacting each other
-Step function is defined, this function performs single transition along an edge in FSM diagram, in definition of step function it is called iteratively over blocks of samples, this process given simply not mentioned how to deal with errors, or where to resume if long sequence of blocks of sample preempted for some reason
-It shows how to make FSM design more optimal, by reducing parameters α and K, Paper claims step bundling method increases efficiency, but no detailed example is given
-Another optimization is proposed as returning additional g_flag variable for step function; but, the affect of it on cost function or time complexity is not discussed

Minor:
Some experiment results are shared on jax. Although jax is popular, pytorch or tensorflow could be used to extend derivation of their approach in different frameworks and libraries

**Questions For Authors:**

see the points listed under "weaknesses" above.

**Relation To Broader Scientific Literature:**

-

**Theoretical Claims:**

No theoretical claims in the paper.

---

> ### Author Rebuttal · Authors · 2025-03-31
>
> Thank you for taking the time to review our work and for your recommendations to improve the paper. Below we address your comments and questions.
>
> **More general while loop structures:**  The conversion of algorithms into FSMs by splitting up while loops is a generally applicable recipe, but we agree that our presentation in Sec 3 sweeps this under the rug by immediately launching into specific algorithms. We have added a paragraph at the outset that explains how to obtain FSMs for any finite number of while loops. This can in fact be done automatically: We have a parser that reads the MCMC algorithm in a symbolic language and outputs the FSM (see below for details). This is already implemented and tested. We had not included it to save space and because all MCMC algorithms with while loops we are aware off fall into the categories considered in our paper. However, since the automatic procedure exactly addresses your point, we will add it to the appendix if the reviewers have no objections.
>
> *General FSM construction procedure:* We have derived an algorithm able to convert arbitrary programs (written in a symbolic programming language) into their respective FSM graph. The algorithm (1) calls a parser to obtain the syntax tree of its input program, (2)  creates a coarsened version of that tree where each leaf corresponds to a code block (and each non-terminal node corresponds to a while loop condition), and (3) recursively transforms that tree into an FSM graph. Each node and edge of the final FSM graph can be mapped back to a code block or a while loop condition of the original program.
>
> **Step function errors:** Our method transforms an existing MCMC program into an equivalent sampler using the same lines of code within blocks. As such, any error handling in the original program is retained by the FSM. To handle preemption, one can straightforwardly checkpoint the state of the FSM and all flags at a desired frequency of `step` updates.
>
> **Bundling method:** In Section 5.1, we provide an illustrative example of `bundled_step` (see Algorithm 4) and discuss how it improves performance over `step`. We also tested the performance of `bundled_step` in the Delayed-Rejection experiment 7.1 (see the discussion under Experimental Setup, Results, and in the caption of Fig 6). However, we agree that more explanation of the effect of step-bundling would improve the paper. To that end, we propose to add text similar to the following in Section 5.1:
>
> "As long as the switch in `step` executes each state/block sequentially when using `vmap` (and our testing finds that it does$^*$), then, both `vmap(step)` and `vmap(bundled_step)` have the same execution cost. This is because `vmap(bundled_step)` always executes all blocks/states sequentially, and both `vmap(bundled_step)` and `vmap(step)` execute all blocks. As a result,  if each chain makes on average M transitions when calling `vmap(bundled_step)`, the speed-up over using `vmap(step)` is M-fold. Since $M \geq 1$ (i.e. `bundled_step` must make at least one transition each time it is called), this means that `bundled_step` must always weakly improve the performance of `step`, with `vmap`".
>
> $^*$We have tested the JAX's `switch` and found that it behaves consistently with sequential block execution. We can add this to the appendix, if desired.
>
> **Amortization method:** We explain in Section 5.2 that amortization ensures any function $g$ that is called in multiple states/blocks, will only be executed once per FSM step. This reduces $\alpha$. In Section 7.2 we also test the effect of amortization on the performance of the elliptical slice sampler (see line 376 onwards).  However, we appreciate that more detail on the effect of amortization would improve the clarity of our paper. We therefore propose to add text similar to the following in Section 5.1:
>
> "If a function $g$ is called in M different states/blocks and its executions cost (in total) $\beta \in [0,1]$ fraction of the cost of all blocks/states  $S_1,...,S_K$, then amortization will reduce the cost of calling `vmap(step)` to $\beta/M + (1-\beta)$ fraction of the original cost, since $g$ is now executed only once, instead of $M$ times. We indeed observe this for the elliptical slice sampler in Section 7.1 (see Fig 5 - here $M=2$ and $\beta \approx 1$ as the log-likelihood is very expensive, resulting in a 2x speed-up over no amortization)".
>
> **JAX vs. Torch/TensorFlow:** We agree that testing other frameworks is of interest. That we have not done so yet is because (1) PyTorch is not applicable, since its version of `vmap`does not yet handle data-dependent control-flow, and (2) TensorFlow’s `vectorized_map` is so similar to JAX’s `vmap` (it is a tensor-level vectorization tool that constructs a single batched while loop for all chains) that we expect it to behave very similarly. We do plan to test this properly in the future, but with limited space, we feel that a proper evaluation for one framework is more helpful.

---

### Official Review · Reviewer_akMN · 2025-03-13

**Overall Recommendation:** 4

**Summary:**

This work proposes a way of more effectively parallelising (multiple chains of) certain MCMC algorithms via automatic vectorisation tools like the vmap function in JAX. Specifically, this work is concerned with MCMC algorithms whose computational cost per iteration has a large variance, e.g., due to the use of a proposal based around rejection sampling.

The authors show that a naive parallelisation of such chains (i.e. by passing functions that execute an entire MCMC iteration to vmap) is wasteful because it leaves most of the processors idle while waiting for the slowest iteration to complete. Instead, they propose to break MCMC iterations down into smaller steps (formalised via finite state machines). This allows automatic vectorisation to occur on the level of these smaller steps rather than on the level of entire MCMC iterations.

**Claims And Evidence:**

The claims are supported by clear and convincing evidence. The computational cost improvement over "naive" vectorisation (on the MCMC iteration level) is established theoretically. Run-time improvements are demonstrated empirically on a number of benchmark examples.

**Essential References Not Discussed:**

None come to mind.

**Experimental Designs Or Analyses:**

I only checked the experiments insofar as checking their description in the manuscript. I did not find any isues.

**Methods And Evaluation Criteria:**

The proposed methods make sense.

**Other Comments Or Suggestions:**

- Figures should not appear in the text on pages before they have been referenced (see, e.g., Fig. 1 which is only referenced on Page 3; or Fig. 6)
- Line 204, RHS: the double use of the index $k$ in the lower bound for $\alpha$ is confusing
- Section 4, 2nd paragraph. Maybe denote the index of the block containing the loop by $k^*$ rather than $k$ to avoid confusing this "special" value of $k$ and $k$ as a generic index below
- Line 2020: define what $c$ with "negated $k$" in the subscript means
- Eq. 10: Is this really equal for finite $n$ (not just approximately equal)?
- Line 284: use used
- Fig. 7: "and or than half" in the caption
- Table 1: use natbib's citet rather than citep for the reference in the caption
- Bibliography: fix missing capital letters in names/proper nouns in article titles, and in journal names; use of journal name abbreviations is also not consistent.

**Other Strengths And Weaknesses:**

I think this work is overall well written and well structured. As far as I know, the contribution is original. In my opinion, the contribution is also significant enough for publication in ICML because it provides a fairly simple way of speeding up multiple-chain MCMC sampling (for certain MCMC kernels) using existing general-purpose automatic vectorisation functionality.

**Questions For Authors:**

Can you give some more guidance for how to optimise the "bundling" for the "bundled_step" routine? Presumably, one has to be careful here because its performance can end up being worse than "step" (i.e. without any bundling).

**Relation To Broader Scientific Literature:**

Effectively using parallel architectures to speed up MCMC implementations is an important research area. This work contributes to it by providing a fairly simple solution to exploit existing general-purpose automatic vectorisation routines.

**Theoretical Claims:**

I checked the main manuscript but not the additional derivations in the appendix. I did not find any issues.

---

> ### Author Rebuttal · Authors · 2025-03-31
>
> Thank you for your detailed review and recommendations. We are glad that you see the value of our contribution and your feedback has been helpful in strengthening the clarity of the paper. Below we address your main questions/comments:
>
> **On step bundling**:  In short, performance should always be weakly improved (i.e. not get worse), since at worst `bundled_step` takes a single step during each call.
>
> *In more detail*:
> - As long as the switch in `vmap(step)` executes the blocks/states sequentially (and our testing indicates that it does), the step-bundling routine should always weakly outperform the basic `step` function. Like `vmap(step)`, `vmap(bundled_step)` will execute all blocks/states (sequentially) when called. However, `bundled_step` allows any chains that transition from $S_{k} \to S_{k+1}$, to (for example) immediately transition from $S_{k+1} \to S_{k+2}$ at no extra cost (assuming $S_{k+1}$ is called after $S_k$  in the step bundling order). As such, in the worst case each chain only progresses through exactly one transition (which mirrors the behaviour of `step`).
> - Whilst one can optimise the ordering of `bundled_step` to improve performance, a surprisingly effective heuristic is just to use the chronological order of $S_1,...,S_K$, since (for all algorithms we considered in this work) $S_{k+1}$ is constructed as a block that is callable after $S_k$ in the original `sample` function. This ordering is what we use in the paper. In our experiments, we have not found a significant gain to manual tuning of the block ordering over chronological ordering.
>
> **Definition of $c_{\neg k}$**: Thank you for pointing out this omission, we meant to define $c_{\neg k}(m) = \sum_{j \neq k} c_j(m)$.
>
> **Eq(10)**: Equation 10 holds asymptotically as $n \to \infty$, as it is derived by plugging in the limits of $C_0(m,n)$ and $C_F(m,n)$, which are given by Theorem 4.1. Since the approximation error for each of these terms is (with a high probability) $\mathcal O(1/\sqrt{n})$, after some basic manipulations (and assuming $C_F(m,n) > 0$) this results in the approximation error $C_0(m,n)/C_F(m,n) - E(m) = \mathcal O(1/\sqrt{n})$ (again, with some fixed high probability $1-\delta$). If space permits and there are no objections, we are happy to add this analysis in the main text.
>
> **Other typos+ formatting suggestions**: Thank you for pointing out the typos, formatting and improvements, we will amend those accordingly in the final version.

---

### Decision · Program_Chairs · 2025-05-01

**Decision:**

Accept (spotlight poster)

**Comment:**

The paper proposes a novel and efficient parallelization scheme for MCMC algorithms. A known limitation of the ${\tt vmap}$ function is its inefficiency when the function to be parallelized includes one or more while loops, as execution is delayed until all parallel processes have completed. The authors address this issue by leveraging finite state machines (FSMs). They demonstrate how common MCMC algorithms involving while loops can be represented using FSMs, and show that combining this representation with ${\tt vmap}$ enables more efficient parallel execution. Their claims are supported by both a theoretical complexity analysis and empirical results.

Most reviewers agree that this is an interesting and valuable contribution to the community. I share this view and recommend acceptance.

## Remark on Theorem 4.1
In the proof of Theorem 4.1, the authors assume that the Markov chain under consideration admits an absolute spectral gap. This assumption should be explicitly stated in the theorem, and a clear definition of the absolute spectral gap should be included in the supplementary material.